# Cross-link assisted spatial proteomics to map sub-organelle proteomes and membrane protein topologies

Ying Zhu [1], Kerem Can Akkaya [1,2], Julia Ruta [1], Nanako Yokoyama[1], Cong Wang[1], Max Ruwolt [1], Diogo Borges Lima[1], Martin Lehmann [2] & Fan Liu [1,3] ✉

The functions of cellular organelles and sub-compartments depend on their protein content, which can be characterized by spatial proteomics approaches. However, many spatial proteomics methods are limited in their ability to resolve organellar sub-compartments, profile multiple sub-compartments in parallel, and/or characterize membrane-associated proteomes. Here, we develop a cross-link assisted spatial proteomics (CLASP) strategy that addresses these shortcomings. Using human mitochondria as a model system, we show that CLASP can elucidate spatial proteomes of all mitochondrial sub-compartments and provide topological insight into the mitochondrial membrane proteome. Biochemical and imaging-based follow-up studies confirm that CLASP allows discovering mitochondria-associated proteins and revising previous protein sub-compartment localization and membrane topology data. We also validate the CLASP concept in synaptic vesicles, demonstrating its applicability to different sub-cellular compartments. This study extends the scope of cross-linking mass spectrometry beyond protein structure and interaction analysis towards spatial proteomics, and establishes a method for concomitant profiling of sub-organelle and membrane proteomes.

Cellular processes are mediated through complex interactions of biological molecules. To precisely control these interactions, cells are compartmentalized into various membrane-bound and membrane-less compartments that carry out specialized functions. Understanding the molecular basis of compartment-specific cellular functions requires insights into the spatial distribution of proteins and their dynamics. To enable protein localization profiling with high throughput and in a system-wide manner, various liquid chromatography mass spectrometry (LC-MS)-based spatial proteomics methods have been developed[1,2], including Dynamic Organellar Maps (DOMs)[3], Localisation of Organelle Proteins by Isotope Tagging (LOPIT)[4], and Protein Correlation Profiling[5,6]. However, these methods depend on sub-

cellular fractionation, which limits their spatial resolution because separation of different organelles is often incomplete and organelle sub-compartments cannot be resolved.

Information on sub-compartment-specific protein localization can be obtained by proximity-dependent enzymatic labeling approaches such as APEX, BioID and similar strategies (TurboID, APEX2, etc.)[7,8]. These methods (referred to as APEX/BioID for the remainder of the paper) rely on fusing a biotinylating enzyme to a protein of known localization or a peptide sequence targeted to a specific sub-compartment, enabling the enzyme-assisted biotin labeling of proximal proteins. Therefore, APEX/BioID methods require multiple experiments to capture different sub-compartments and applying them to

[1]Department of Structural Biology, Leibniz-Forschungsinstitut für Molekulare Pharmakologie (FMP), Robert-Roessle-Str. 10 13125, Berlin, Germany. [2]Department of Molecular Physiology and Cell Biology, Leibniz-Forschungsinstitut für Molekulare Pharmakologie (FMP), Robert-Roessle-Str. 10 13125, Berlin, Germany. [3]Charité – Universitätsmedizin Berlin, Charitépl. 1, 10117 Berlin, Germany. ✉e-mail: fliu@fmp-berlin.de

characterize membrane-associated proteomes remains challenging. Furthermore, their labeling radius is difficult to control[9], compromising the spatial resolution. In addition, the required engineering and ectopic expression of the target protein/peptide may introduce artifacts.

A potential yet unexplored alternative to fractionation- and proximity labeling-based spatial proteomics methods is cross-linking mass spectrometry (XL-MS)[10]. In XL-MS, proteins are covalently linked using small organic molecules (cross-linkers) composed of a spacer arm and two functional groups that are reactive toward specific residues. Subsequently, LC-MS is used to identify residue-to-residue cross-links. A cross-link can only occur if the distance between two residues is small enough to be bridged by the cross-linker. Consequently, the radius of XL-MS is clearly defined by the spacer arm length of the selected cross-linker[11], which is typically 5–20 Å (0.5–2 nm). This suggests that XL-MS may enable spatial proteome profiling at a higher and more easily controllable spatial resolution than BioID (ca. 10 nm[12]), APEX (ca. 20 nm[13] and 269 ± 41 nm[9]), and μMap-based labeling (ca. 4 nm[14] and 54 ± 12 nm[9], currently only applicable to cell surface proteins). However, even though we and others have developed methods for proteome-wide XL-MS[10,15,16] and have shown that these approaches can capture large parts of the proteome in intact cells and organelles[17–23], cross-linking has so far only been used to analyze protein structures and interactions.

Here, we demonstrate that, beyond its utility in structural biology and interactomics, cross-linking enables high-resolution systematic mapping of protein localizations and membrane protein topologies. We establish the concept of cross-link assisted spatial proteomics (CLASP) by analyzing intact human mitochondria cross-linked with the commonly used disuccinimidyl sulfoxide (DSSO)[24] reagent. Mitochondria of cultured human cells are a popular model system for establishing spatial proteomics methods[25–27] because they consist of multiple spatially distinct sub-compartments - the outer membrane (OMM), the intermembrane space (IMS), the inner membrane (IMM), and the matrix[28]. They have been extensively characterized by APEX/BioID methods[26,27,29,30], which are the only other approaches that can yield sub-compartment spatial information and insights into membrane protein topology. These published APEX/BioID datasets can serve as a performance benchmark for CLASP. Collectively, the mitochondrial model system offers architectural features and published data that allow us to evaluate whether CLASP can (1) determine protein localization with sub-compartment resolution, (2) characterize several sub-compartments in parallel, (3) capture membrane protein topology, and (4) add value compared to existing spatial proteomics approaches. To show that CLASP is broadly valid and utilizable, we apply it to another human mitochondria dataset and to a dataset obtained with synaptic vesicles (SVs) from mouse brain, both generated using the enrichable azide-tagged, acid-cleavable disuccinimidyl bis-sulfoxide (hereafter referred to as DSBSO)[31] cross-linker. While the CLASP analysis of DSBSO-cross-linked SVs serves as an additional proof of concept, CLASP of DSBSO-cross-linked mitochondria deepens the coverage of the mitochondrial interactome, allowing us to confirm most of the spatial annotations from our original DSSO-based CLASP analysis and add more localization and topology predictions. We verify several of our findings through biochemical, imaging and bioinformatics approaches, demonstrating the effectiveness and robustness of CLASP for elucidating protein localizations with high spatial resolution.

## Results

### Proteome-wide XL-MS can reveal a high number of inter-protein connections

CLASP is based on the idea that system-wide XL-MS experiments are very likely to capture some well-characterized proteins with known subcellular localization, since such localization markers (LMs) tend to be highly abundant. The cross-links of these LMs allow deducing the relative localization of the directly connected proteins. CLASP analysis can thus extract spatial information from any XL-MS dataset of a biological system, provided that (1) a substantial number of inter-protein cross-links is detected, (2) cross-links are formed with a defined labeling radius, and (3) some LMs are captured. To assess whether these three conditions are fulfilled in a standard proteome-wide XL-MS experiment, we analyzed intact mitochondria isolated from HEK293T cells using DSSO cross-linker in three biological replicates (Fig. 1A, Supplementary Fig. 1). We identified 13971 unique cross-links from 1451 proteins by imposing a 2% FDR at the level of unique cross-linked residue pairs separately for intra-protein and inter-protein links. This dataset includes 6250 intra-protein and 7721 inter-protein cross-links corresponding to 2606 protein-protein connections (Supplementary Data 1–4), confirming that our XL-MS dataset meets the first requirement for CLASP.

### Cross-linking provides a well-defined maximum labeling radius

To test whether our dataset also meets the second criterion, we set out to define the labeling radius of the applied DSSO cross-linker. It has a spacer arm of 10.3 Å and connects lysine residues, which have a side chain length of 7.6 Å. Taking into account a protein in-solution flexibility of 10 Å, we hypothesized that the maximum labeling radius in our CLASP experiment (defined as the $C_\alpha$-$C_\alpha$ distance between two cross-linked lysines) will be < 4 nm/40 Å.

Traditional XL-MS studies have firmly established that the maximum cross-linked $C_\alpha$-$C_\alpha$ distances are in good agreement with high-resolution protein structures. The cross-links in our mitochondrial dataset confirm this: We mapped 483 cross-links (357 intra-protein links and 126 inter-protein links) on high-resolution structures of 31 mitochondrial protein complexes and found that 96% of the measured Cα-Cα distances agree with the 4 nm distance constraint (Supplementary Figs. 2A and S3). However, this type of validation is limited to structurally characterized protein complexes, still raising the question whether the CLASP labeling radius is equally well controlled for the remaining proteins.

To begin to address this point, we took advantage of existing knowledge on the mitochondrial ultrastructure. Comparing our XL-based network to sub-cellular localization information from Swiss-Prot shows that we detected proteins in all mitochondrial sub-compartments, proving that the cross-linker penetrated both the OMM and the IMM. The distance between OMM and IMM is 20 nm as previously shown by electron tomography[28]. Therefore, an OMM protein should only cross-link to an IMM protein if their IMS regions are long enough to reduce the distance between their lysines to <4 nm. We analyzed this possibility for the only two OMM-IMM cross-links we detected between LMs: ATAD3A-TOMM20 and ATAD3A-TOMM40. Considering the linear length of their IMS regions (calculated as 3.5 Å per amino acid * number of amino acids), both cross-linked lysine pairs fall into the permitted distance (Supplementary Fig. 2B). Notably, we do not observe OMM-IMM cross-links between proteins whose IMS protrusions are too short to achieve a mutual distance of 4 nm or lower.

As another line of evidence, we analyzed cross-links involving mitochondrial membrane proteins of known topology. Although DSSO penetrates both OMM and IMM, we should not observe cross-links across these membranes because their thickness is about 7 nm[32], exceeding the assumed maximum labeling radius for DSSO. In agreement with this assumption, we do not detect any cross-links across the OMM and IMM, i.e. all our cross-linked amino acid pairs are localized within the same membrane-bound sub-compartment. To illustrate this, we show that the IMM-embedded proteins of the mitochondrial respiratory chain are exclusively involved in membrane-separated, sub-compartment-specific cross-links (Supplementary Fig. 2C). This finding provides further evidence that the DSSO labeling distance is well-controlled and does not exceed 4 nm, thus fulfilling the second requirement for CLASP.

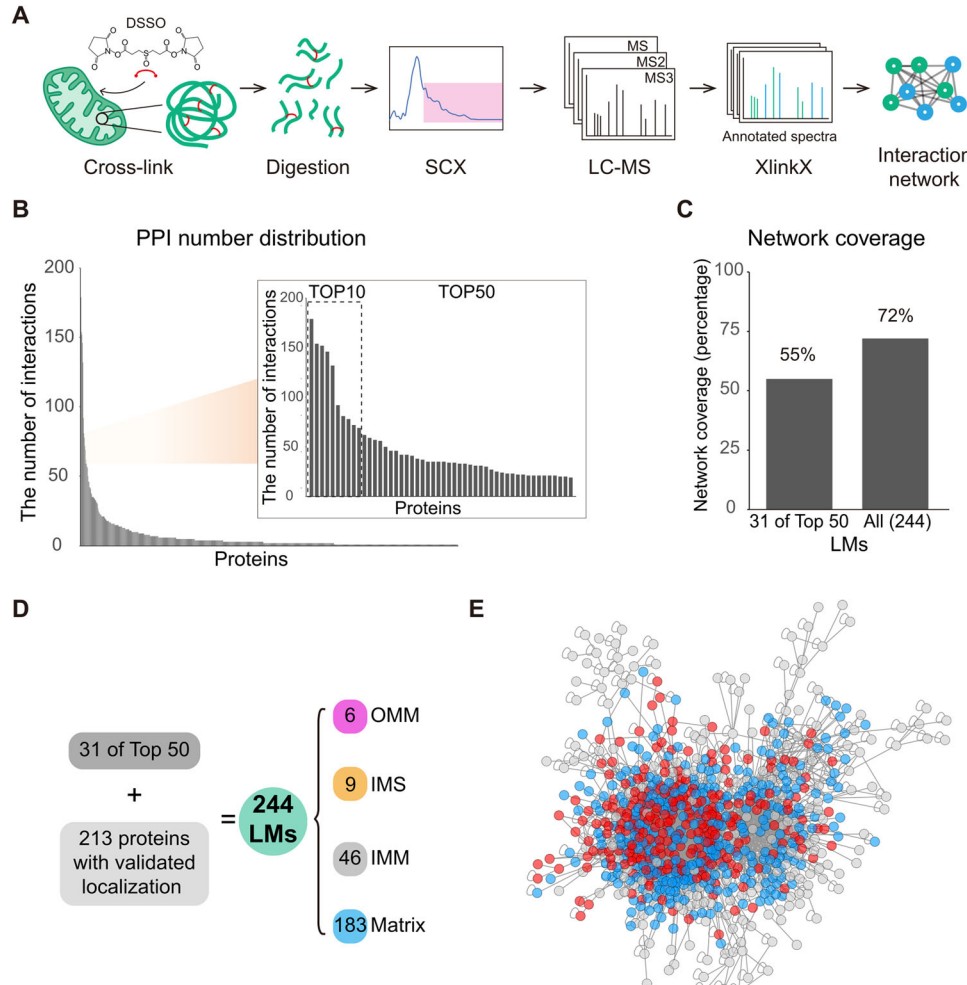

**Fig. 1 | CLASP reveals localization information for most proteins in a XL-based mitochondrial PPI network. A** Workflow for the XL-MS analysis of human mitochondrial proteins. **B** All identified proteins ranked by the number of their interactions. The inset shows the top 10 and top 50 most-connected proteins. The top 10 proteins are HSPD1, MDH2, HSPE1, ATP5F1A, HSPA9, C1QBP, CYCS, ATP5F1B, PHB2, and SHMT2. **C** Network coverage achieved when considering the first-tier interactors of 31 LMs derived from the top 50 most-connected proteins (left bar) and all LMs (right bar). **D** The origins and sub-compartment localizations for all 244 LMs. Proteins were selected as LMs if (i) their sub-mitochondrial localization had been

thoroughly established in previous work, (ii) they were part of the top 50 most-connected proteins, had a corresponding PDB structure, or were a component of a well-studied mitochondrial protein assembly, and (iii) there are no cross-links contradicting their sub-compartment localization. **E** Overview of the mitochondrial PPI network, showing that XL-MS achieves a high degree of interconnectivity. The majority of the network is covered by LMs (red) and their first-tier interactors (blue), i.e. the proteins considered in the CLASP analysis. Source data are provided as a Source Data file.

## XL-MS captures reliable localization markers for CLASP-based spatial predictions

To evaluate whether our standard XL-MS experiment also fulfilled the third prerequisite for CLASP – the detection of robust LMs – we first filtered our XL-MS dataset by removing proteins lacking inter-protein links or forming small disconnected clusters resulting in an interactome of 2523 protein-protein connections among 748 proteins. Since these proteins are part of a well-connected interactome and thus in direct contact to other mitochondrial or mitochondria-associated proteins, they can be assigned to this organelle with reasonable confidence. To put this 748-protein dataset into context, we compared it to published mitochondrial proteome resources. In particular, we considered the MitoCarta3.0 database[33], the MitoCoP database generated by combining different proteomics approaches[34] and mitochondrial datasets generated by DOMs[35], LOPIT-DC and HyperLOPIT[36] (Supplementary Fig. 4). This analysis reveals a core of 286 proteins, which appear in the DSSO XL-MS data as well as all published datasets and databases. It also shows that 73% of the mitochondrial proteins found by DSSO cross-linking have been assigned to this organelle by at

least one other resource. This overlap is similar to the results of the DOMs dataset, which show a 72% agreement with other resources.

To identify LMs in this DSSO-cross-linking-based mitochondrial interactome, we assessed the protein connectivity (Fig. 1B). We found that 40 of the 50 most connected proteins have well-established sub-mitochondrial localizations. Since our goal is to use CLASP for making biological discoveries with the highest possible confidence, we sought to maximize the number of reliable LMs in our mitochondrial protein network by also including several well-characterized mitochondrial proteins and complexes such as VDAC, TIM/TOM complex, mitochondrial contact site and cristae organizing system (MICOS), oxidative phosphorylation complexes (electron transport chain complexes I-IV and ATP synthase) and the mitochondrial ribosome. To assess their suitability as LMs, we curated spatial information from the published literature and high-resolution structures (Supplementary Data 5), which led to the identification of another 218 LM candidates. To assess the robustness of the spatial information delivered by all LM candidates, we extracted the sub-network of these proteins and evaluated it for consistency. We deemed information as conflicting whenever the

cross-links of a protein led to unreconcilable sub-compartment annotations. Among 4502 cross-links, 12 contradictory cross-linked residue pairs were found, meaning that 99.7 % of cross-links among LM candidates are internally consistent and support the known LM localizations. The 12 cross-links providing contradictory information might be caused by proteins with multiple sub-compartmental localizations or false-positive cross-link identifications. After removing LM candidates with conflicting cross-link information, 244 high-confidence LMs remained, including 31 of the 50 most connected proteins and 213 well-characterized mitochondrial proteins (Fig. 1D and Supplementary Data 5). Importantly, the first-tier interactors of these LMs (i.e. proteins connected through direct cross-links) cover 72% of the XL-based network, or 55% when only considering LMs that are among the 50 most abundant proteins (Fig. 1C, E). This confirms that XL-MS readily captures well-connected LMs that make the majority of the detected interactome amenable to CLASP localization annotations. Importantly, these annotations span all mitochondrial sub-compartments, showing that CLASP allows profiling multiple sub-compartments in parallel and determining protein localizations with sub-organelle resolution.

Having shown that the faithful capture of LMs is feasible in mitochondria, we wanted to assess whether this concept holds true when applying a different XL-MS workflow and studying another organelle. We selected synaptic vesicles (SVs) isolated from mouse brain. SVs are single membrane-bound compartments for the storage of neurotransmitters. The constitutive SV proteome is small – a previous study of rat SVs suggested that less than 50 proteins are present with at least one copy number per compartment[37] – and essential SV functions are mediated by integral or associated membrane proteins[38,39]. Many of these have a well-established localization/topology and thus are well suited to independently test our LM concept.

Whereas mitochondrial samples were prepared using DSSO cross-linking and cross-link enrichment by strong cation exchange chromatography, mouse brain SVs were cross-linked using the membrane-permeable, enrichable, and MS-cleavable DSBSO cross-linker[40], and subjected to click-chemistry-based enrichment and subsequent size exclusion chromatography. We identified 5456 cross-links from mouse brain SVs by imposing a 2% FDR at the level of unique cross-linked residue pairs. The dataset includes 417 protein-protein connections (Supplementary Data 6). We focused on 28 proteins with established SV membrane localization/topology (Supplementary Data 7), qualifying as LMs according to the criteria we established above. Among these LMs, we identified 181 intra-protein and 84 inter-protein cross-links corresponding to 36 protein-protein connections. Our data showed 100% agreement with the known localization/topology of the 28 LMs (Supplementary Fig. 5). This confirms that XL-MS data are very likely to yield internally consistent LMs, and thus meet all criteria for performing CLASP.

## CLASP facilitates efficient protein localization mapping

After validating the CLASP concept, we analyzed the CLASP predictions for DSSO-cross-linked mitochondria in more detail. We found that 91% of the CLASP annotations within our dataset are unambiguous, meaning that all LMs support the same subcellular localization. These 91% include 49% (146 proteins) confirming previous reports, 3% (9 proteins) contradicting published localization data, and 39% (115 proteins) presenting spatial and topological information for previously unannotated proteins (Fig. 2A). CLASP suggests sub-compartment localizations for 97 hitherto unmapped proteins, revises the topology of 24 membrane proteins (see section "CLASP characterizes the topologies of membrane proteins" below), and discovers 3 mitochondria-associated proteins.

To further evaluate the scope of CLASP, we compared our dataset to two published BioID datasets generated in a similar system: the HEK293 cell map[7] and the HEK293 mitochondrial proximity interactome[29] (Fig. 2B, C).

The HEK293 cell map determined mitochondrial localizations of 503 proteins by two data analysis algorithms (see Methods), but annotations are limited to three categories (matrix, IMM/IMS and OMM/peroxisome), indicating that the spatial resolution is too low to fully discern all sub-compartments. Our CLASP experiments provided localizations for fewer proteins (270) but resolved sub-compartment localizations in more detail (see Supplementary Data 5, column 'Sub-cellular location after CLASP annotation'). This allowed us, for example, to identify specific proteins for each individual sub-compartment and distinguish between proteins protruding into different adjacent sub-compartments, e.g. matrix-facing vs. IMS-facing IMM proteins. A similar spatial granularity was reported by Antonicka et al. who also captured the highest number of mitochondrial proteins (1465) (Fig. 2B)[29]. However, the fraction of single sub-compartment annotations was lower in the Antonicka et al. dataset (69%) than in our CLASP dataset (91%, Fig. 2C). Although proteins with more than one locale within mitochondria are possible, previous work suggests that less than 20% of the mitochondrial proteins may be dually localized[41]. This indicates that the high fraction of unambiguous sub-compartment annotations by CLASP is an accurate depiction of mitochondrial biology and a result of CLASP's higher spatial specificity. In line with this, 158 proteins of our CLASP dataset were also identified by Antonicka et al. but 84 of them were ambiguously annotated. Reassuringly, of the 74 unambiguous protein annotations by both CLASP and Antonicka et al., 80% are consistent.

Furthermore, both BioID datasets are based on assays with 12[7] and 100[29] ectopically expressed baits from all mitochondrial sub-compartments, compared to three CLASP experiments in native mitochondria. This demonstrates that – by obviating exogenous fusion proteins and instead taking advantage of well-characterized mitochondrial proteins to determine protein localizations – CLASP substantially increases the yield of spatial information per experiment (Fig. 2D).

## CLASP reveals biologically relevant sub-compartment localizations

Having evaluated the fundamental features of CLASP, we next assessed its potential for biological discovery. We determined sub-compartment localizations for 209 non-membrane proteins, 97 of which were previously unassigned. Of these, 23 proteins were mapped to the IMS and 74 to the matrix (Supplementary Data 5).

We selected FAM136A for complementary validation. Confocal imaging of rat tissue[42] and multiple human cell lines (see https://www.proteinatlas.org/ENSG00000035141-FAM136A/subcellular#human) has shown that endogenous FAM136A resides in mitochondria. Mito-Carta3.0 places FAM136A in the IMS, based on APEX studies that found an enrichment of this protein in the IMS compared to the cytosol but did not consider other mitochondrial sub-compartments[30] Therefore, FAM136A's sub-compartment localization has not been definitively characterized. Our CLASP analysis, which covered all mitochondrial sub-compartments, indeed suggests that FAM136A localizes to the IMS (Fig. 3A). We verified this annotation by confocal and STED microscopy, showing that FAM136A co-localizes with the OMM marker TOMM20 and occurs within mitochondria (Fig. 3B, C). Alkaline carbonate extraction (Fig. 3D) and a protease protection assay (Fig. 3E) further confirmed that FAM136A is not membrane-bound and resides in the IMS. Collectively, these data provide comprehensive evidence that FAM136A locates to the IMS in human cells.

As a second example, we followed up on NIPSNAP2/GBAS. This protein locates to the mitochondrial matrix according to MitoCarta3.0 and Abudu et al.[43] whereas Swiss-Prot (O75323, last reviewed on January 19th, 2024) and Shanmughapriya et al.[44] annotate it as an OMM protein. According to CLASP, NIPSNAP2 resides in the matrix (Supplementary Fig. 6A) and this is confirmed by protease protection assays (Supplementary Fig. 6B). We additionally verified NIPSNAP2's

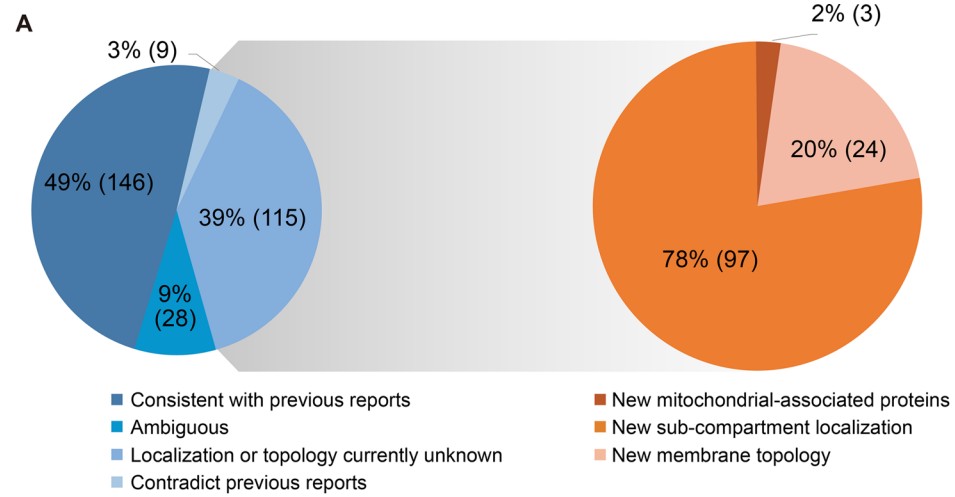

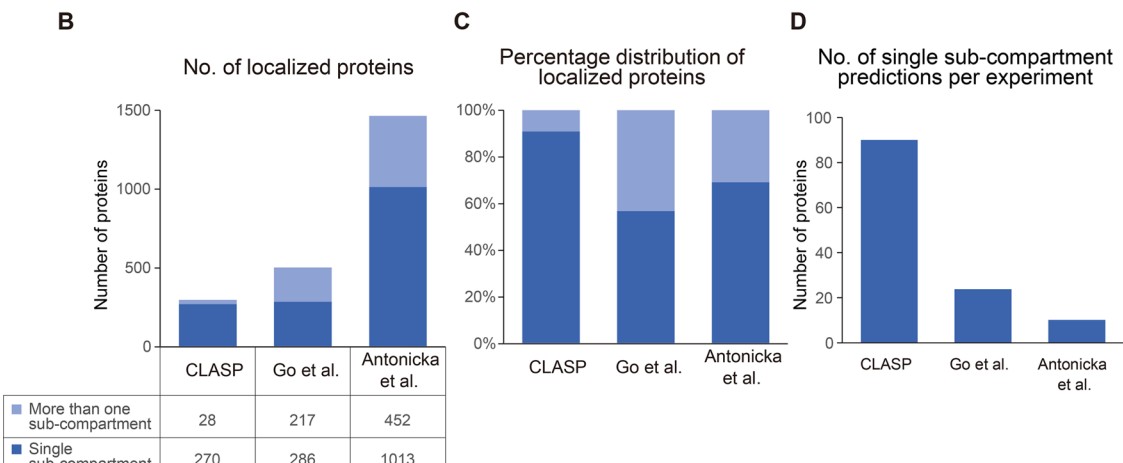

**Fig. 2 | Evaluation of CLASP performance. A** Comparison of CLASP annotations to published protein localization information (blue pie chart, left) and breakdown of CLASP annotations that disagree with previous reports or relate to previously unannotated proteins (orange pie chart, right). Annotations of individual proteins are shown in Supplementary Data 5. **B**, **C** Number (**B**) and percentage distribution (**C**) of mitochondrial protein localization annotations by CLASP and in published proximity biotinylation resources[7,29]. The columns are sub-divided to indicate the spatial specificity of the localization annotations. **D** Estimated information yield of a single CLASP experiment and single BioID experiments in[7] and[29], calculated as the total number of single sub-compartment annotations divided by the total number of assays. Source data are provided as a Source Data file.

location by selective permeabilization experiments, in which mitochondria are treated with digitonin (permeabilizes only OMM) or triton (permeabilizes OMM and IMM). Subsequent confocal imaging reveals under which detergent conditions the proteins become accessible for the detection antibodies. OMM and IMS proteins become readily detectable upon digitonin treatment, whereas matrix proteins can only be detected in the presence of triton. These experiments show that NIPSNAP2 behaves similarly to the matrix marker SDHA and clearly different from the OMM marker TOMM20 (Supplementary Fig. 7). This demonstrates that NIPSNAP2 is a matrix protein and not an OMM protein in our experimental setting.

### CLASP discovers mitochondria-associated proteins
We found 3 proteins, which have no previously reported association with mitochondria, cross-linked to the cytosolic side of OMM LMs (Supplementary Data 5). Considering the DSSO labeling radius of 4 nm, we hypothesized that these proteins may directly bind to the OMM. One of them, FAF2 (UBXD8), is known as an endoplasmic reticulum (ER) membrane protein involved in ER-associated degradation[45]. We found FAF2 directly connected to the cytosolic parts of the OMM LM

VADC2 and to the ER- and mitochondria-localized protein CYB5R3[46] (Fig. 4A), suggesting that FAF2 might localize to both mitochondrial OMM and the ER. This is supported by a protease protection assay showing a similar digestion pattern for FAF2 and the OMM protein TOMM20 (Fig. 4B) and thus supporting the CLASP annotation of FAF2 as an OMM protein with a cytosol-facing C-terminus. Additionally, FAF2 co-localizes with TOMM20 and the ER marker Calreticulin according to confocal imaging, corroborating its localization at both organelles (Fig. 4C–E). After we had made this discovery (published on bioRxiv in May 2022), an independent study confirmed that FAF2 (UBXD8) localizes to the ER and mitochondria in human cells[47] and a subsequent study suggested that this protein is involved in regulating mitochondria-ER contact sites[48]. These complementary findings showcase the potential of CLASP as a discovery tool that can guide functional follow-up experiments.

### CLASP characterizes the topologies of membrane proteins
In the previous sections, we established that CLASP enables high-resolution spatial profiling owing to its clearly defined labeling radius. Another factor contributing to its high resolution is that CLASP

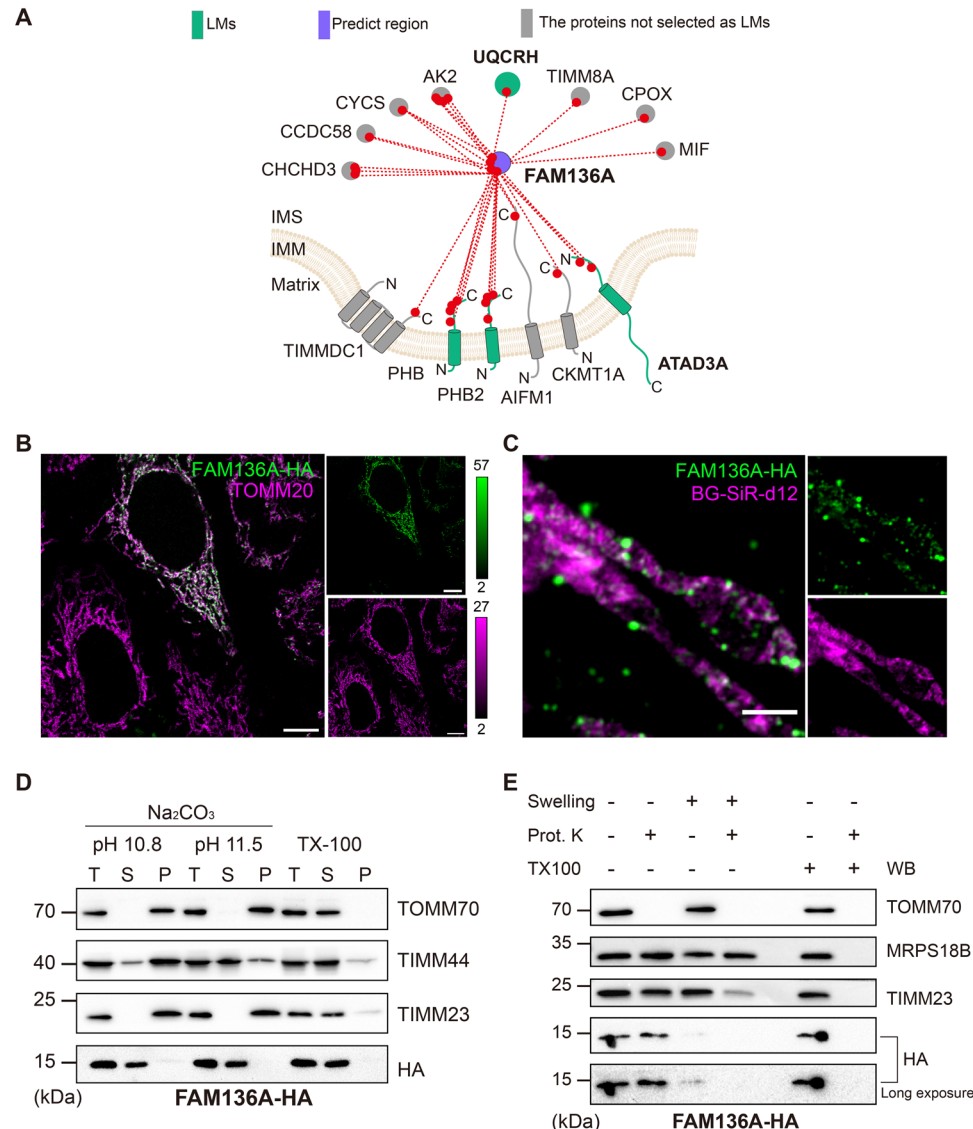

**Fig. 3 | CLASP confidently localizes FAM136A to the IMS. A** Cross-link map of FAM136A and its interacting proteins. LMs are shown in green; FAM136A is shown in purple. The CLASP annotation of FAM136A is supported by direct connections to 1 IMS LM and 3 IMM LMs. Proteins that are not LMs but support the IMS localization of FAM136A are shown in grey. "Predict region" indicates the protein/protein region, for which a CLASP prediction was made. **B** Confocal fluorescence images of C-terminally HA-tagged FAM136A (FAM136A-HA) and OMM marker TOMM20 in HeLa cells (left panel). Images of untransfected cells are shown as a negative control in the bottom left. The color bar shows the contrast setting. Scale bars = 10 μm. **C** STED microscopy images of FAM136A-HA-transfected HeLa-COX8A-SNAP cells stained with BG-SiR-d12 (to label the IMM marker COX8A) and HA antibody. Scale bar = 1 μm. **D.** Alkaline carbonate extraction of mitochondria isolated from HEK293T cells overexpressing FAM136A-HA. The OMM protein TOMM70, IMM protein TIMM23 and IMM associated protein TIMM44 are used as markers for each sub-compartment. T, total mitochondrial extraction; S, supernatant; P, pellet of mitochondrial membrane. **E** Protease protection assay to analyze the localization of FAM136A-HA in HEK293T cells. OMM protein TOMM70, IMM protein TIMM23 and matrix protein MRPS18B are used as markers for each mitochondrial sub-compartment. Experiments in panels B-E were performed once. Source data are provided as a Source Data file.

provides sub-compartment localization information on the residue level. In other words, CLASP can distinguish regions within one protein that face different sub-compartments. This offers the opportunity to use CLASP for assessing the topology of membrane proteins. In our dataset, we annotated 58 mitochondrial membrane proteins, for which we could assign membrane regions based on previous findings and computational predictions (see Methods for details). The topological annotations of 34 proteins are consistent with previous studies. For the remaining 24 proteins, we were able to revise or complete their topological annotation based on CLASP data (Supplementary Data 5 and Fig. 5).

We sought to validate two CLASP predictions: CYB5R3 as an OMM protein with a cytosol-facing C-terminus and CYB5B as an OMM protein with an IMS-facing C-terminus. Protease protection assays,

confirm both of these CLASP results (Supplementary Fig. 8). Furthermore, we found that CLASP can help correct existing topology annotations: Swiss-Prot annotates TMEM126A as an IMM protein with matrix-facing N- and C-termini but CLASP predicts that the TMEM126A termini protrude into the IMS (Fig. 6A). TMEM126A has a similar digestion pattern as COX8A-SNAP, which has an IMS-facing C-terminus (Fig. 6B), again supporting the CLASP result (Fig. 6A, right panel). We additionally confirmed the IMM localization of TMEM126A through a selective permeabilization assay (Supplementary Fig. 7). Supporting our previous results, TMEM126A shows the highest similarity to the IMM markers TIMM23 and FAM162A. Furthermore, it shows a higher similarity to OMM marker TOMM20 than to matrix marker SDHA, confirming that its termini (the regions that are bound by the detection antibody) are facing the IMS.

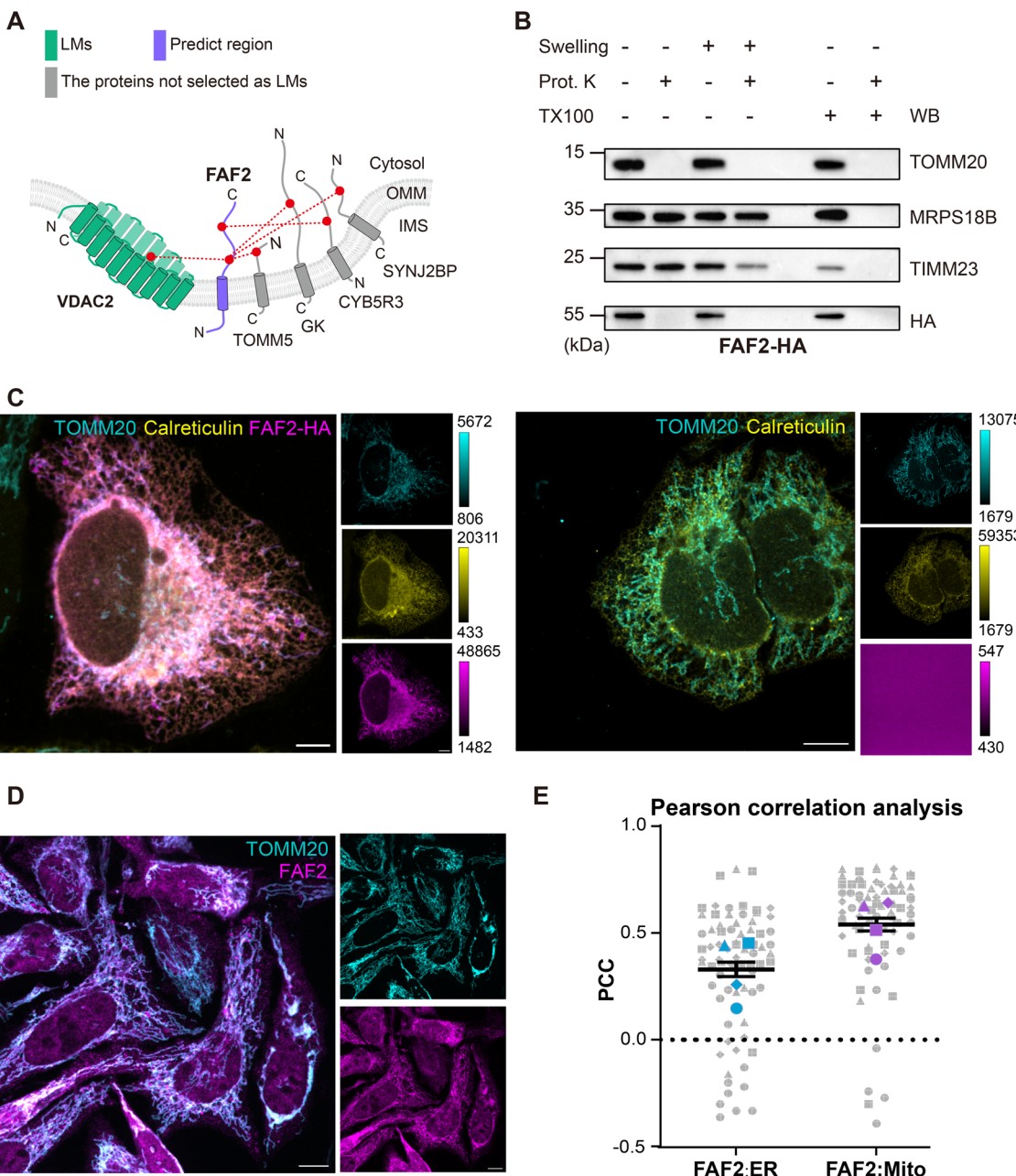

**Fig. 4 | CLASP discovers FAF2 as a mitochondria-associated protein. A** Cross-link map of FAF2 and its interacting proteins. LMs are shown in green; FAF2 is shown in purple; proteins that are not LMs but supporting the CLASP annotation of FAF2 are shown in grey. "Predict region" indicates the protein/protein region, for which a CLASP prediction was made. **B** Protease protection assay to analyze the localization of HA-tagged FAF2 (FAF2-HA) in HEK293T cells. OMM protein TOMM70, IMM protein TIMM23 and matrix protein MRPS18B are used as markers for each mitochondrial sub-compartment. **C** Confocal fluorescence images of FAF2-HA, Calreticulin (ER marker) and TOMM20 (mitochondria marker) in HeLa cells (left panel). Untransfected negative control image for FAF2-HA staining are displayed in the right panel. Shown are representative images from a total of 4 independent experiments. The color bar shows the contrast setting. Scale bars = 10 μm. **D** Confocal fluorescence images of endogenous FAF2 (magenta) and TOMM20 (cyan) in HeLa cells. Scale bars = 10 μm. **E** Pearson correlation analysis of the co-localization of FAF2 with ER and mitochondria markers from panel C. Shown are mean values +/− SEM for n = 4 independent experiments with multiple cells per experiment. Data from each independent experiment are depicted with a distinct symbol type. Colored symbols indicate the means of the respective experiments. Experiments in panels (**B**, **D**) were performed once. Source data are provided as a Source Data file.

To put these CLASP topology data in context, we compared them to a proximity biotinylation study[26] that used an APEX approach to predict IMM protein topologies in a similar HEK293 system. Lee et al. designed three APEX fusion proteins (and several more for validation experiments) to obtain membrane topology information for 60 IMM proteins, whereas CLASP provided topological insights into 42 IMM proteins from native mitochondria (Supplementary Fig. 9). Of the 17 protein topologies determined by both CLASP and APEX, 14 (82%) are consistent. Furthermore, 43 APEX annotations are missing in the CLASP dataset. Most of these proteins (24) have well established topologies, which is why they served as LMs in CLASP. Of the remaining APEX-only proteins, 12 were not detected by CLASP, 6 were not annotated because they did not directly cross-link to any LM, and 1 annotation was ambiguous (Supplementary Fig. 9). Conversely, CLASP gave topology information for 25 IMM and 16 OMM proteins not included in the APEX

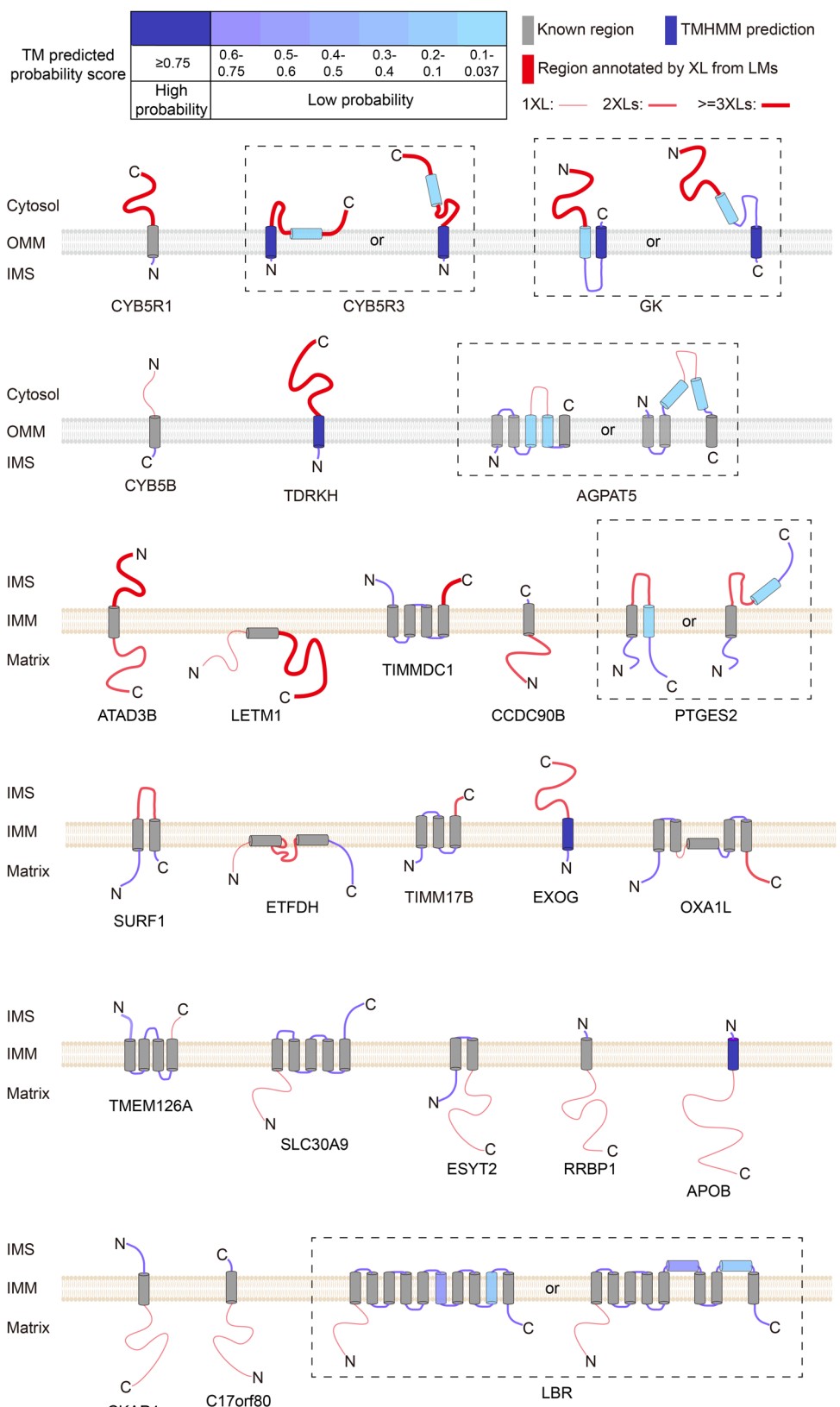

**Fig. 5 | CLASP determines the membrane topologies of 24 OMM and IMM proteins.** The TM regions predicted by TMHMM2.0 are gradient colored according to their posterior probabilities of TM helix; Swiss-Prot annotated TM regions are shown in grey; the regions annotated by CLASP are shown in red, the soluble regions predicted by TMHMM2.0 are shown in shades of blue. Of note, TMHMM2.0 cannot predict the localization of the soluble regions, but CLASP allows determining the orientation of the TM regions and localization of the predicted soluble regions. Dashed boxes indicate different possible topologies of one protein. Previously known and predicted TMs are included in Supplementary Data 5.

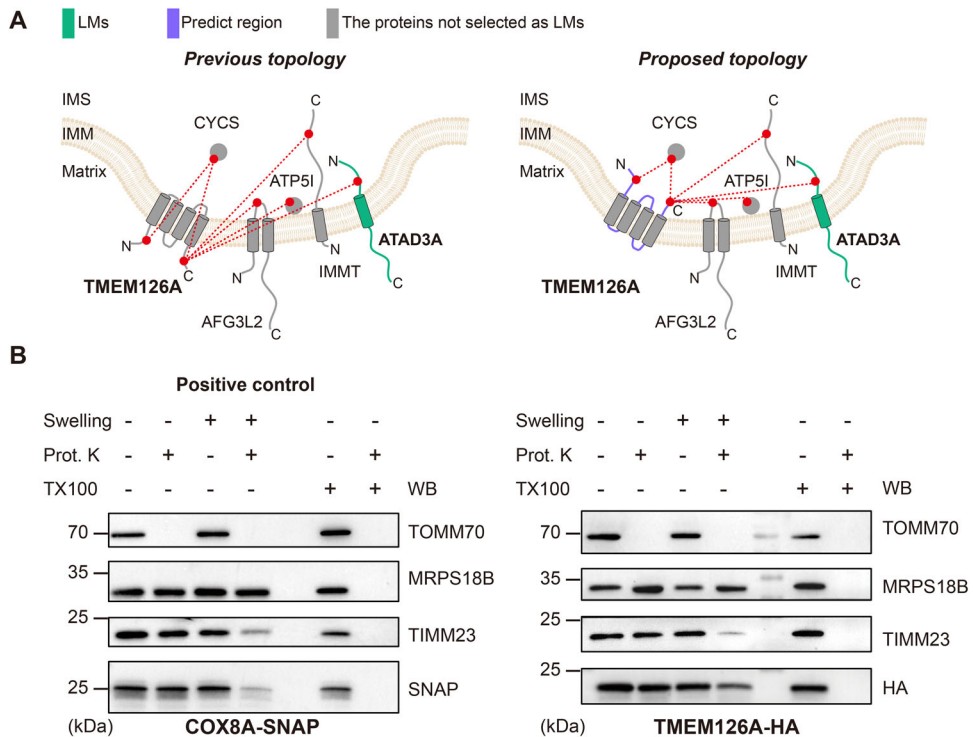

**Fig. 6 | CLASP corrects the topology annotation for TMEM126A. A** Cross-link map of TMEM126A and its interacting proteins. Left: the cross-link map based on the previous annotation; right: the cross-link map based on the CLASP annotation. LMs are shown in green; predicted regions by TMHMM2.0 are shown in purple. IMS-localization of the TMEM126A C-terminus is supported by 1 LM. Since TMEM126A has an even number of TM regions, both termini are likely located in the same sub-compartment, which is confirmed by additional connections to proteins

that are not LMs (shown in grey). "Predict region" indicates the protein/protein region, for which a CLASP prediction was made. **B** Protease protection assay combined with WB to analyze the localization of TMEM126A-HA in HEK293T cells. COXA8-SNAP, which has the same membrane topology as the one CLASP suggested for TMEM126A, serves as a positive control. Experiment was performed once. Source data are provided as a Source Data file.

dataset, in addition to capturing numerous sub-compartment localizations as described above.

## CLASP with an enrichable cross-linker validates and deepens spatial insights into mitochondria

Having verified several CLASP predictions through biochemical and imaging assays, we next wanted to assess whether these findings can be reproduced with a different cross-linker. Therefore, we performed CLASP on a mitochondrial XL-MS dataset generated with the MS-cleavable and enrichable DSBSO cross-linker (Supplementary Fig. 10A). In addition, DSBSO gives us the opportunity to test whether using an enrichable cross-linker can augment CLASP.

We took the same data analysis approach as for the DSSO data: Imposing a 2% FDR (separately for inter-protein links and intra-protein links) at the level of unique cross-linked residue pairs yielded 26650 unique cross-links from 2041 proteins, including 11426 intra-protein and 15224 inter-protein cross-links corresponding to 5163 protein-protein connections (Supplementary Data 9). After removing proteins lacking inter-protein links or forming small disconnected clusters, we obtained a DSBSO-interactome comprising 5094 protein-protein connections (doubling the number of the DSSO interactome) among 1205 proteins (a 61% increase compared to the DSSO interactome).

For LM selection, we again focused on proteins with well-characterized sub-mitochondrial localizations, prioritizing those that are among the most connected proteins in the DSBSO interactome. The LM candidates formed 7209 cross-links amongst each other, 21 of which were contradictory to their known locations. Thus, 99.7% of cross-links among LM candidates are internally consistent and support the known LM localizations. After removing LM candidates with

conflicting cross-link information, 235 high-confidence LMs remained (Supplementary Fig. 10B, Supplementary Data 10).

Focusing on the first-tier interactors of these LMs (i.e., proteins connected through direct cross-links), DSBSO-based CLASP yielded 420 predictions, adding or correcting the topology of 30 membrane proteins as well as the sub-mitochondrial localizations of 122 soluble proteins (Supplementary Fig. 10C, Supplementary Data 10). Reassuringly, DSBSO-based CLASP captured 431 out of 542 spatially annotated proteins from the original DSSO dataset (Supplementary Fig. 11)

Mirroring our original findings with DSSO, the DSBSO-based CLASP localization predictions are highly specific with 93% of the results being unambiguous (i.e. all LMs support the same sub-compartment localization). Overall, 163 proteins were unambiguously localized by both DSSO-based and DSBSO-based CLASP (Supplementary Fig. 12 A). These overlapping predictions showed an 98% agreement (Supplementary Fig. 12B), further validating the robustness and reproducibility of CLASP.

At the same time, the greater depth of the DSBSO dataset allowed to predict the localizations of several additional proteins. We selected four of these additional predictions (Fig. 7A) for complementary validation by protease protection and alkaline carbonate extraction assays (Fig. 7B–E). Specifically, we focused on

- CNP (2′,3′-cyclic-nucleotide 3′-phosphodiesterase), an established mitochondrial protein known to regulate the mitochondrial permeability transition pore during cell death[49,50], which has not been assigned to a sub-compartment so far and is CLASP-annotated as a matrix protein.
- MIF (Macrophage migration inhibitory factor), a secretable cytoplasmic protein, which has been shown to control the regulation

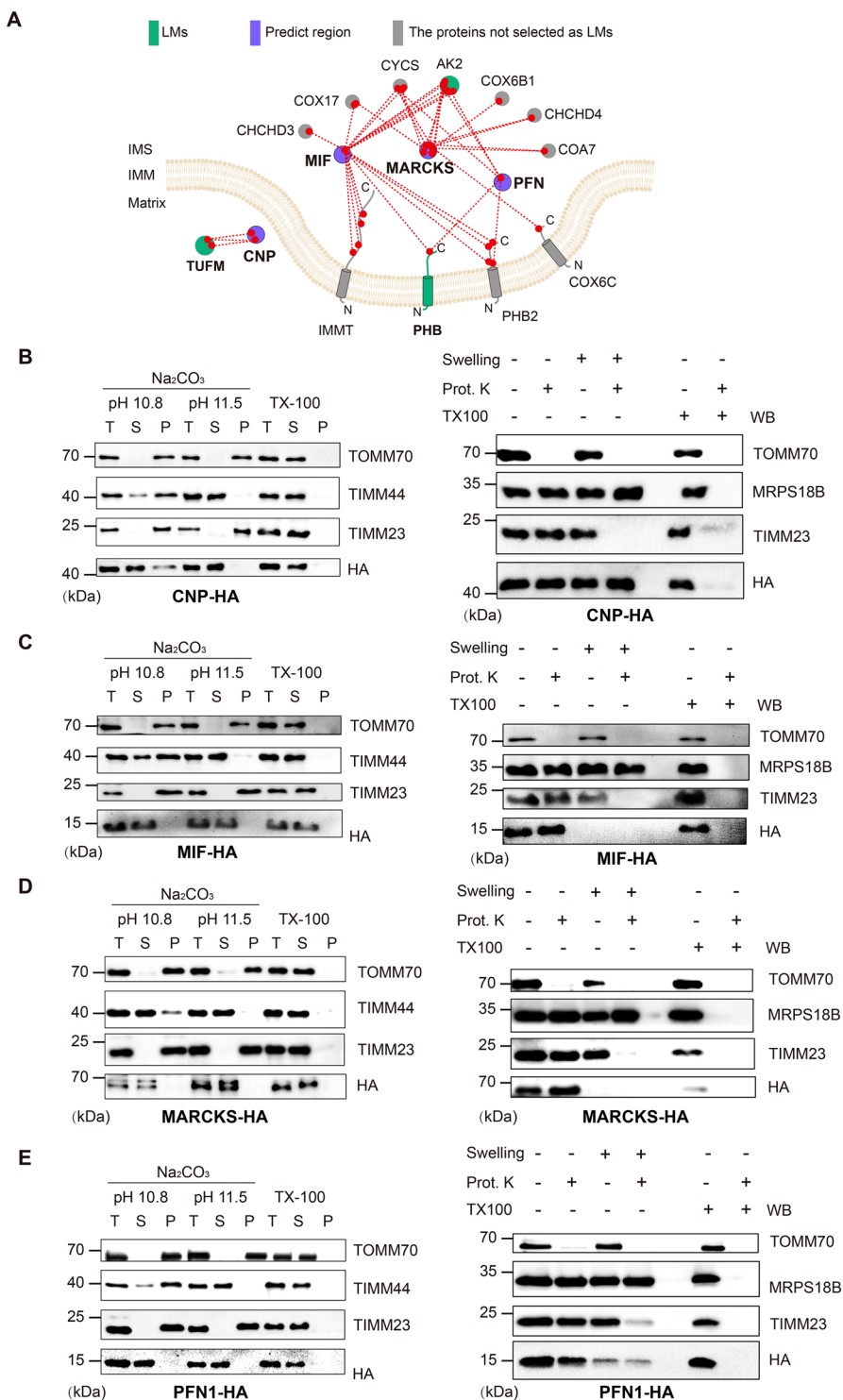

**Fig. 7 | DSBSO-based CLASP reveals matrix localization of CNP matrix protein and IMS localization of MIF, MARCKS and PFN1. A** Cross-link map of CNP, MIF, MARCKS, PFN1 and their interacting proteins based on DSBSO cross-linking. LMs are shown in green; The predicted proteins are shown in purple. Proteins that are not LMs but support the predicted localization are shown in grey. "Predict region" indicates the protein/protein region, for which a CLASP prediction was made. **B–E** (left panel) Alkaline carbonate extraction of mitochondria isolated from HEK293T cells overexpressing CNP-HA, MIF-HA, MARCKS-HA, PFN1-HA separately.

The OMM protein TOMM70, IMM protein TIMM23 and IMM associated protein TIMM44 are used as markers for each sub-compartment. T, total mitochondrial extraction; S, supernatant; P, pellet of mitochondrial membrane. **B–E** (right panel) Protease protection assays to analyze the localization of CNP-HA, MIF-HA, MARCKS-HA, PFN1-HA in HEK293T cells separately. OMM protein TOMM70, IMM protein TIMM23 and matrix protein MRPS18B are used as markers for each mitochondrial sub-compartment. Experiments in (**B–E**) were performed once. Source data are provided as a Source Data file.

- mitochondrial dynamics by averting apoptosis[51]. CLASP predicts that MIF resides in the IMS.
- MARCKS (myristoylated alanine-rich C-kinase substrate), which was previously found to bind various cellular membranes including the OMM and be able to dissociate into the cytoplasm[52]. By contrast, CLASP localizes MARCKS to the IMS.
- PFN (Profilin-1), which was recently suggested to be critical for mitochondria respiration, morphology, and dynamics, and to localize to the matrix[53]. However, CLASP predicts an IMS localization for PFN.

In all four cases, protease protection and alkaline carbonate extraction assays confirmed that the proteins are not membrane bound and reside in the sub-compartment predicted by CLASP (Fig. 7B–E). That said, it is important to bear in mind that protein locations may change upon certain molecular and environmental triggers. For example, the localization of MARCKS has previously been shown to depend on post-translational modifications[52] Additionally, proteins may reside in more than one compartment and their dominant location may vary. This might be the case for PFN, as its band pattern in our protease protection assay (Fig. 7E) suggests IMS as the main locale but does not fully exclude a second localization in the matrix. More generally, we cannot exclude that the localizations found here in HEK293 cells may differ from the localizations in other biological systems or conditions. Nonetheless, our biochemical data further support the accuracy of the CLASP predictions.

### A Python workflow for automated CLASP predictions

Finally, we sought to increase the general accessibility of CLASP by developing a Python tool that automates LM selection and spatial predictions. The tool only requires an XL-MS data table and protein localization annotations from Swiss-Prot as input. First, it will automatically select LMs based on the Swiss-Prot annotations. Second, it will predict protein localizations and membrane topologies by combining cross-link information and defined LMs. The pipeline gives users the opportunity to manually add more LMs as this may increase the confidence of CLASP predictions. In this regard, we found that protein abundance is a useful metric to identify additional LM candidates, since more abundant proteins not only are more likely to be biologically well characterized but also tend to have a higher number of cross-linking-partners, i.e. proteins for which CLASP predictions can be made (Supplementary Fig. 13).

We used our DSSO dataset to compare the tool-based CLASP predictions (Supplementary Data 11) to our original CLASP results (Supplementary Data 5). Reassuringly, feeding all our manually selected LMs into the Python tool results in predictions that are 100% identical to our original DSSO-based CLASP results, confirming the functionality of our software. When relying on the tool-based automated LM selection, 98.8% of cross-links among the candidate LMs agreed with their reported locations. After removing LMs with conflicting cross-link information, we obtained 72 LMs, which yielded almost only unambiguous localization predictions (95%). The vast majority of the automatically obtained LMs and first-tier interactors have also been found using our original manual approach (298 out of 343, Supplementary Fig. 14). For 98% (292 proteins) of the overlapping identifications, automated and manual CLASP give the same annotation of sub-compartment localization and membrane topology. Overall, these results demonstrate that the developed Python tool offers an easy and reliable option for performing CLASP.

## Discussion

Over the last two decades, XL-MS has become an established tool in structural biology, complementing methods such as X-ray crystallography and cryoEM with low-resolution structural information on purified proteins and protein complexes in solution. Recent methodological advancements further expanded the scope of XL-MS to more complex biological systems[10,15], but XL-MS applications remained focused on the structural analysis of proteins and discovery of protein interactions. In this study, we have developed a framework that broadens the application of XL-MS to spatial proteome profiling. Using purified human mitochondria as a model system, our CLASP approach allowed us to confirm existing spatial annotations for 146 proteins and add localization and topology data for 124 proteins (Fig. 2A and Supplementary Data 5). The scope of CLASP is most comparable to proximity labeling-based spatial proteomics methods, in particular APEX/BioID-derived strategies. However, a look at previous studies reveals important differences.

First, several proximity labeling-based spatial proteomics studies also took advantage of the unique morphology of mitochondria and used it as a model system for method benchmarking. These studies demonstrated, for example, the capabilities of APEX-based proximity labeling to reveal protein sub-compartment localization[27,30], membrane protein topology[26], and membrane contact sites[25]. Characterizing each of these spatial features required the design of dedicated proximity labeling experiments, whereas we have shown that CLASP can provide insights into all these aspects simultaneously in one experiment and from native mitochondria. Accordingly, CLASP substantially increases the yield of spatial information per experiment (Fig. 2D). The main reason for this fundamental difference is that APEX/BioID methods require genetic engineering of fusion proteins targeted to a specific localization. While this strategy has undisputed advantages for profiling cell type- or localization-specific proteomes in organisms[54–56], it takes substantial efforts to generate different constructs and to validate the correct localizations of the fusion proteins. Furthermore, the fusion proteins need to be ectopically expressed, which may affect the cellular status and thereby the spatial distribution of the endogenous proteins-of-interest.

Second, APEX/BioID as well as all other spatial proteomics methods derive spatial information from quantitative proteomics data and therefore must rely on statistical methods to determine which proteins are significantly enriched in one location vs another. By contrast, CLASP does not require quantitative information because it derives spatial predictions directly from qualitative XL-MS data (i.e. cross-link identifications that pass the FDR cutoff and other filtering criteria). That said, CLASP can, in principle, be applied to any comprehensive proteome-wide XL-MS dataset from an intact biological system and quantitative proteome-wide XL-MS may be used in the future to study changes of protein localizations.

A third important difference between CLASP and APEX/BioID approaches is the spatial resolution (Supplementary Fig. 15). The BioID resolution is limited to ca. 10 nm[12]. While there have been attempts to reduce the APEX labeling radius[57], the free diffusion of APEX-generated radicals remains difficult to control. This may be a reason for the vast differences in estimated APEX labeling radii (ca. (20 nm[13] and $269 \pm 41$ nm[9]). The comparably large labeling radii in APEX/BioID potentially compromise their utility for the highly selective labeling of specific cellular microenvironments. In support of this notion, we provide evidence that, compared to existing proximity labeling resources, CLASP yields a larger fraction of specific localization annotations (Fig. 2C).

The localization confidence of CLASP is further increased by its ability to consider multiple LMs. This feature allows cross-validating spatial annotations through LMs with overlapping interactors and verifying the fidelity and coherence of the LMs themselves. Furthermore, we have shown that LM selection in CLASP is automatable through computational approaches. While the Python tool we developed fully relies on Swiss-Prot information that may be augmented by user input, we envision that future developments may result in more sophisticated software tools that can predict the most powerful LM

combination based on the architecture of the XL-based network and existing localization evidence.

The reliance of CLASP on LMs also means that its predictive power will be lower for biological systems with fewer well-characterized LMs. However, decades of spatial biology research have generated a wealth of protein localization information, yielding hundreds of potential LMs particularly for the most relevant biological systems in fundamental and clinical research. CLASP is ideally suited to fill the remaining blank spots in the spatial protein maps of these systems because it can take advantage of existing localization data to locate previously unmapped proteins. At the same time, CLASP is still applicable to systems with sparse endogenous LMs. CLASP annotations in such systems could be improved by ectopically expressing tagged constructs as exogenous LMs, which would be conceptually more similar to the APEX/BioID fusion protein approach but still offer the benefits of CLASP's higher spatial resolution.

Our benchmarking of CLASP omitted an analysis of its temporal resolution, which will chiefly depend on the choice of cross-linker. The DSSO and DSBSO cross-linking reactions performed in this study usually require at least 15 min incubation time during which proteins may still be partially mobile. Other cross-linking chemistries with reaction times under 30 s have been published recently[58] and may offer additional options for temporally resolved CLASP in the future. Similar developments have occurred in the proximity labeling field, where alternative labeling methods promise a better temporal control than traditional APEX/BioID approaches[59,60].

We demonstrated the general applicability of CLASP by applying it to two sub-cellular compartments with distinct features: mitochondria and SVs. In addition, we have recently shown that the proximity information derived from XL-MS data can be used to distinguish functionally and spatially distinct protein layers within intact viral particles, which are not separated by any physical barrier[61]. Considering that CLASP is based on the same type of data and can use XL-MS datasets of any spatially defined biological system as input, these findings strongly suggest that CLASP will also be applicable to membrane-less compartments. At the same time, it is important to note that the fidelity of CLASP annotations critically depends on the comprehensiveness, accuracy, and interconnectivity of the XL-based protein network. Such detailed protein networks can be generated for purified organelles by state-of-the-art XL-MS workflows[17–20], as also shown by the detection depth achieved in this study. Therefore, CLASP can readily provide detailed localization annotations for purifiable organelles, provided that sufficient input material is available. CLASP would likely be less powerful when applied to limited sample amounts (e.g. primary samples) or intact cells, since proteome coverage of most in-cell XL-MS workflows is currently still limited. However, recent technological advancements of XL-MS have brought the identification of tens of thousands of cross-links in intact cells within reach[22,23]. CLASP paves the way to use these data for elucidating protein localizations across the cell.

## Methods

### Plasmid construction

Total RNA was isolated from HEK293T cells (ATCC) by TRIzol reagent (Invitrogen) according to the manufacturer's instruction. cDNA library was obtained by using First Strand cDNA Synthesis Kit (Thermo Fisher). For FAF2-HA and CYB5R3-HA, the open reading frames without stop codon were PCR-amplified by Phusion™ High-Fidelity DNA Polymerase (Thermo Fisher) using the forward and reverse primer pairs. The PCR product was cloned into the HindIII and BamHI sites of pcDNA3.1(+) plasmid by using GeneArt™ Seamless Cloning and Assembly Enzyme Mix (Invitrogen). OMP25-HA was cloned into the HindIII and BamHI sites of the pSNAP-N1 plasmid. pSNAPf-Cox8A Control Plasmid was a gift from New England Biolabs & Ana Egana (Addgene plasmid # 101129). The primers used for cloning were

synthesized by BioTeZ Berlin-Buch GmbH and are listed in Supplementary Data 8. All other plasmids were purchased from Absea Biotechnology Ltd. All plasmids used in this study were verified by DNA sequencing.

### Cell culture and transfection

HEK293T cells (ATCC) were cultured in Dulbecco's Modified Eagle Medium (DMEM) supplemented with 10% fetal bovine serum at 37 °C and 10% $CO_2$. Plasmid transfections were performed using Lipofectamine 2000 (Invitrogen) according to the manufacturer's instruction. Briefly, plasmid DNA and Lipofectamine 2000 were mixed with Opti-MEM separately at a ratio of 3:1 of Lipo2000 to plasmid DNA, and then Lipo2000 was added into the plasmid DNA immediately. The mixture was incubated for 20 min and added dropwise to the cell. After 48 h transfection, cells were washed and harvested with ice-cold PBS for further experiments.

### Mitochondria isolation

Mitochondria isolation was modified from previous published protocols[62] as follows: Three hundred million HEK293T cells were resuspended in ice-cold buffer M (220 mM mannitol, 70 mM sucrose, 5 mM HEPES-KOH pH 7.4, 1 mM EGTA-KOH pH 7.4, supplemented with 1 mM PMSF and complete protease inhibitor EDTA-free cocktail). Cells were lysed by homogenization (25 strokes, 2 times, 900 × rpm) using dounce homogenizer. Cell debris were spun down at 800 × g for 5 min at 4 °C 2 times. The supernatants were centrifuged at 10,000 × g for 10 min at 4 °C and the pellet was collected. The pellet containing crudely purified mitochondria was further subjected to discontinuous percoll gradient centrifugation (SW41 Ti rotor, Beckman) to obtained high purity mitochondria. Protein concentration was determined by Bradford assay (Bio-rad). The crudely purified mitochondria were used for protease protection assay and alkaline carbonate extraction experiment. High purity mitochondria were used for the cross-linking experiment.

### Synaptic vesicle (SV) purification

SVs were purified from C57BL/6 mouse brain as previously described[63,64]. The mouse work complied with all relevant ethical requirements. All animal experiments were reviewed and approved by the ethics committee of the "Landesamt für Gesundheit und Soziales" (LAGeSo) Berlin and were conducted according to the committee's guidelines under animal experimentation permits. At the institute, the Animal Care Officer and the LAGeSo monitored compliance with all regulations. The mice were looked after by professional caretakers and checked daily. Mice were kept in an animal facility that is regularly checked for standard pathogens. Mice were kept in groups of up to six animals in standard individually ventilated cages of 524 cm² at (21 +/ −2) °C, (50 +/−10)% relative humidity, and a 12:12 h light−dark cycle. Cages contained bedding and nesting material. Food and water were provided ad libitum.

Briefly, 10 brains from 6-week-old male and female mice were homogenized by dounce homogenizer (9 strokes, 900 × rpm) in ice-cold homogenization buffer (320 mM sucrose, 4 mM HEPES-KOH pH 7.4, supplemented with 1 mM PMSF and complete protease inhibitor EDTA-free cocktail). Cell debris were spun down at 800 × g for 10 min at 4 °C. The supernatants were centrifuged at 10,000×g for 15 min at 4 °C 2 times and the pellet (synaptosome) was collected. The pellet was osmotically lysed by ice-cold water and homogenization (3 strokes, 2000×rpm). Plasma membranes were spun down at 25,000 × g for 20 min at 4 °C (SW41 Ti rotor, Beckman). The supernatants were centrifuged at 200,000 × g for 2 h at 4 °C. The pellet (enriched SVs) was collected and used for the cross-linking experiment.

Integrity of the SVs in the preparations was assessed by negative-stain electron microscopy (Supplementary Fig. 16). To this end, the pellet was resuspended in homogenization buffer and diluted in

phosphate-buffered saline. The sample was applied to a glow-discharged continuous carbon grid and imaged on a Zeiss 910 (FEI Morgagni).

## Protease protection assay

Protease protection assays were performed following previous published protocols[65]. Briefly, freshly isolated mitochondria were suspended in SEM buffer (250 mM sucrose, 10 mM MOPS, 1 mM EDTA, pH 7.2), EM buffer (10 mM MOPS, 1 mM EDTA, pH 7.2) or EM buffer containing 1% Triton X-100 respectively and incubated on ice for 25 min. Proteinase K (PK) was added into the samples and incubated for 10 min on ice. The reaction was quenched by addition of 2 mM PMSF, followed by trichloroacetic acid (TCA) precipitation. After treatment, the pellet was resuspended in SDS sample buffer and subjected to western blot analysis.

## Alkaline carbonate extraction

Alkaline carbonate extraction was performed following previous published[65]. Briefly, freshly isolated mitochondria were suspended in 0.1 M $Na_2CO_3$ pH 11.5, 0.1 M $Na_2CO_3$ pH 10.8 or SEM buffer containing 1% Triton X-100 respectively and incubated on ice for 20 min. The samples were centrifuged at 100,000 × g for 1 h at 4 °C (S55A2 rotor, Thermo Fisher). The pellets were resuspended in SDS sample buffer and subjected to western blot analysis.

## Western blot analysis

Protein samples were subjected to SDS-PAGE (14% gel) and wet-transferred to a 0.2 μm Immobilon-PSQ PVDF membrane (Millipore) at 110 V for 90 min. Blots were blocked with 5% BSA for 1 h at room temperature and incubated with corresponding primary antibodies at 4 °C overnight. The following antibodies were used in this study: anti-HA (1:1000, mouse, Abcam, ab18181); anti-SNAP (1:1000, rabbit, NEB, P9310S); anti-MRPS18B, anti-TIMM44 and TOMM70 (1:500, 1:1000, 1:500, rabbit, ProteinTech, 16139-1-AP, 13859-1-AP, 14528-1-AP); anti-TIMM23 (1:1000, mouse, DB Biotech, 611222). After washing 3 times with TBST, blots were incubated with secondary antibody peroxidase-conjugated affinipure goat anti- mouse IgG and peroxidase-conjugated affinipure goat anti-rabbit IgG (H + L, Jackson ImmunoResearch, 115-035-003, 111-035-144) for 1 h at room temperature. After washing 3 times with TBST, blots were developed with Pierce ECL Western Blotting Substrate (Thermo Fisher) and imaged by ChemiDoc MP Imager (Bio-Rad).

## Cross-linking of mitochondria

The mitochondrial pellet was resuspended to 1 mg/ml in Buffer M and cross-linked by adding two aliquots of 0.5 mM disuccinimidyl sulfoxide (DSSO) or two aliquots of 0.5 mM DSBSO, each for 20 min at room temperature with constant mixing. The reaction was quenched with 20 mM Tris-HCl, pH 8.0 for 30 min at room temperature. Mitochondria were collected by centrifugation at 10000 × g for 10 min at 4 °C. DSSO cross-linking was done in three biological replicates, DSBSO cross-linking was done once.

## Cross-linking of SVs

The SV pellet was resuspended to 1 mg/ml in homogenization buffer and cross-linked with 2 mM DSBSO for 15 min at room temperature with constant mixing. The reaction was quenched with 20 mM Tris-HCl, pH 8.0 for 15 min at room temperature. DSBSO cross-linking of SVs was done once. Of note, the differences in cross-linking conditions between mitochondria and SVs are a consequence of the different properties of the cross-linkers and the samples being cross-linked.

## Protein digestion

Cross-linked mitochondria were digested in solution. Briefly, urea was added to the mitochondrial pellet to reach a final concentration of 8 M.

Proteins were reduced and alkylated with 5 mM DTT (1 h at 37 °C) and 40 mM chloroacetamide (30 min at room temperature in the dark). Proteins were digested with Lys-C at an enzyme-to-protein ratio of 1:75 (w/w) for 4 h at 37 °C. After diluting with 50 mM TEAB to a final concentration of 2 M urea, trypsin was added at an enzyme-to-protein ratio of 1:100 (w/w) for overnight at 37 °C. The digestion was quenched by adding formic acid to a final concentration of 1%. Peptides were desalted with Sep-Pak C18 cartridges (Waters) according to the manufacturer's protocol, dried in a vacuum concentrator.

Cross-linked SVs were precipitated by chloroform-methanol method. Briefly, three sample volumes of water, four sample volumes of methanol and one sample volume of chloroform were added into the sample and mixed vigorously. The sample was centrifuged at 15,000 × g for 10 min at room temperature using a tabletop centrifuge, then the upper layer was carefully removed. Three sample volumes of methanol were added and the sample was centrifuged at 13,000 × g for 10 min. The precipitated cross-linked SV proteins in the pellet were then digested using the same method as described above.

## Enrichment of DSBSO-cross-linked mitochondria and SV peptides

The digested cross-linked mitochondria and SV peptides were enriched by dibenzocyclooctyne (DBCO)-coupled sepharose beads. Briefly, the mitochondrial or SV peptides were resuspended to 3 mg/ml in PBS and were then added to the prewashed DBCO beads for incubation overnight at room temperature with constant mixing. The bead-to-peptide ratio was 20 μl beads (40 μl slurry) per 0.6 mg peptide. After incubation, the beads were washed with water, followed by a washing step with 0.5% SDS at 37 °C for 15 min. Next, the beads were washed thrice with 0.5% SDS, thrice with 8 M urea in 50 mM TEAB, thrice with 10% ACN and twice with water using ten bead volumes each. The cross-linked peptides were eluted with two bead volumes 10 % (v/v) tri-fluoracetic acid (TFA) for 2 h at 25 °C and then dried using a vacuum concentrator.

## Size exclusion chromatography (SEC) fractionation of DSBSO-cross-linked mitochondrial and SV peptides

SEC fractionation was performed on 160 μg peptides using a Super-dexTM 30 Increase 3.2/300 column (GE Healthcare) on an Agilent 1260 Infinity II system. For mitochondria, a 60 min gradient was applied and 24 fractions were collected and dried in a vacuum concentrator. For SVs, a 90 min gradient was applied and 13 fractions were collected and dried in a vacuum concentrator.

## Strong cation exchange (SCX) fractionation of DSSO-cross-linked mitochondrial peptides

SCX fractionation was performed on mitochondrial digested peptides using PolySULFOETHYL-ATM column (100 × 4.6 mm, 3 μm particles, PolyLC INC.) on an Agilent 1260 Infinity II UPLC system. A 95 min gradient was applied and fractions were collected every 30 sec. 50 late SCX fractions were desalted by C18 stageTip and dried in a vacuum concentrator.

## High pH (HPH) fractionation of DSBSO-cross-linked mitochondrial peptides

HPH fractionation was performed on mitochondrial digested peptides after SEC fractionation using Phenomenex Gemini C18 column on an Agilent 1260 Infinity II UPLC system. A 85 min gradient was applied and 42 fractions were collected and dried under SpeedVac.

## LC-MS analysis

For the analysis of DSSO-cross-linked mitochondrial peptides, collected SCX fractions were analyzed by LC-MS using an UltiMate 3000 RSLC nano LC system coupled on-line to an Orbitrap Fusion Lumos mass spectrometer (Thermo Fisher Scientific). Reversed phase

separation was performed with an in-house packed C18 analytical column (Poroshell 120 EC-C18, 2.7 μm, Agilent Technologies). 3 h LC-MS runs were performed for cross-linking acquisition and 2 h runs were done for proteomic analysis. Cross-link acquisition was performed using a CID-MS2-MS3 acquisition method. MS1 and MS2 scans were acquired in the Orbitrap mass analyzer and MS3 scans were acquired in the ion trap mass analyzer. Notably, MS3 acquisitions were only triggered when peak doublets with a specific mass difference ($\Delta = 31.9721$ Da) were detected in the CID -MS2 spectra, as this is indicative for the presence of DSSO cross-linked peptides. The following MS parameters were applied: MS resolution 120,000; MS2 resolution 60,000; charge state 4-8 enable for MS2; MS2 isolation window, 1.6 m/z; MS3 isolation window, 2.5 m/z; MS2- CID normalized collision energy, 25%; and MS3-CID normalized collision energy, 35%. For mitochondrial proteome analysis, mass analysis was performed using an Orbitrap Fusion mass spectrometer (Thermo Fisher Scientific) with a HCD-MS2 acquisition method. MS1 scans were acquired in the Orbitrap mass analyzer and MS2 scans were acquired in the ion trap mass analyzer. The following MS parameters were applied: MS resolution 120,000; MS2 resolution 60,000; charge state 2-4 enable for MS2; MS2 isolation window, 1.6 m/z; MS2- HCD normalized collision energy, 30%.

For the analysis of DSBSO-cross-linked samples, collected SEC fractions (SV peptides) or HPH fractions (mitochondrial peptides) were analyzed by LC-MS using an UltiMate 3000 RSLC nano LC system coupled on-line to an Orbitrap Fusion Lumos mass spectrometer with a FAIMS Pro device (Thermo Fisher Scientific). Reversed phase separation was performed with an in-house packed C18 analytical column (Poroshell 120 EC-C18, 2.7 μm, Agilent Technologies). 3 h LC-MS runs were performed for cross-linking acquisition. Cross-link acquisition was performed using a stepped higher collisional dissociation (stepHCD) MS2 acquisition method. MS1 and MS2 scans were acquired in the Orbitrap mass analyzer. The following MS parameters were applied: MS resolution 120,000; MS2 resolution 60,000; charge state 4-8 enable for MS2; MS2 isolation window, 1.6 m/z; stepped normalized collision energy, 19-25-30% with FAIMS voltages set to −50, −60, and −75.

## MS data analysis

For cross-linking samples, raw data were converted into MGF files in Proteome Discoverer (version 2.4). Data analysis was performed using a stand-alone version of XlinkX[66] with the following parameters: minimum peptide length 6; maximal peptide length 35; missed cleavages 3; fix modification: Cys carbamidomethyl; variable modification: Met oxidation; DSSO cross-linker =158.0038 Da (short arm = 54.0106 Da, long arm = 85.9824 Da); precursor mass tolerance 10 ppm; fragment mass tolerance 20 ppm. MS2 spectra were searched against a reduced target-decoy Swiss-Prot human database derived from proteins combining MitoCarta2.0 database and protein identification from a mitochondrial proteomic measurement. Results were reported at 2% FDR at the level of unique cross-linked residue pairs. Protein-protein interaction (PPI) network was constructed by Cytoscape software (version 3.8.2). For synaptic vesicles sample, XlinkX stand-alone with the following parameters: minimum peptide length 6; maximal peptide length 35; missed cleavages 3; fix modification: Cys carbamidomethyl; variable modification: Met oxidation; DSBSO cross-linker =308.0038 Da (short arm = 54.0106 Da, long arm = 236.0177 Da); precursor mass tolerance 10 ppm; fragment mass tolerance 20 ppm. MS2 spectra were searched against a reduced target-decoy Swiss-Prot mouse database. Results were reported at 1% FDR at CSM level. Protein- protein interaction (PPI) network was constructed by Cytoscape software (version 3.8.2). Proteomics data were analyzed using Max-Quant software (version 1.6.2.6a) with the following searching parameters: precursor mass tolerance 20 ppm, fragment mass tolerance 20 ppm; fixed modification: Cys carbamidomethylation; variable modification: Met oxidation, protein N-term Acetyl; enzymatic digestion: trypsin/P; maximum missed cleavages: 2. Database search was performed using Swiss-Prot database of all human proteins without isoform (retrieved on May 2020, containing 20,365 target sequences); false discovery rate: 1%. Intensity-based absolute quantification (iBAQ) was enabled to determine to relative abundances of proteins.

## Structural validation for cross-links

Cross-links were mapped onto selected high-resolution structures from the Protein Data Bank (PDB) using Pymol 2.1.0 (Schrodinger LLC). PDB accession codes are provided in the Data Availability section. If homologous non-human structures were used, sequences were aligned to the human protein sequence using NCBI BLAST and cross-links were mapped to the aligned residues.

## Confocal immunofluorescence and STED microscopy

For standard IF, HeLa cells (ATCC) and HeLa-COX8A-SNAP cells (reported in ref. [67] and kindly donated by the authors) were seeded on μ-Slide 8 Well Glass Bottom (ibidi, REF 80827) one day before transfection, cultivated in DMEM containing 10 % FBS, 1 % P/S and 1% L-Glut. After 24 h, the transfected cells were fixed with PBS containing 4% paraformaldehyde (PFA) and 4% sucrose for 10 min at room temperature. Afterwards, fixation was quenched by removing PFA and adding PBS containing 0.1 M glycine and 0.1 M NH₄Cl for 10 min at room temperature. Cells were permeabilized by PBS containing 0.15 % TritonX-100 for 10 min at room temperature and washed by PBS two times. Then sequentially incubated in blocking buffer (PBS containing 1% BSA and 6% NGS) for 30 min at room temperature, blocking buffer containing primary antibodies for 1 h at room temperature. After three washes in blocking buffer, the cells were incubated in blocking buffer containing secondary antibody for 30 min at room temperature. After three washes in blocking buffer, the cells were imaged using confocal microscopy.

For STED microscopy using HeLa-COX8A-SNAP, cells were incubated for 1 h with 1 μM BG-SiR d12, followed by three times washing with DMEM medium shortly before fixation. Cells were fixated in culture, by extraction of half of the DMEM medium and adding PBS containing 4% paraformaldehyde (PFA) and 4% sucrose for 20 min at 37 °C. Afterwards, the standard IF protocol was followed.

For selective permeabilization experiments, cells were seeded onto μ-Slide 8 Well Glass Bottom (ibidi, REF 80827). Transient transfection with selected candidate plasmids was performed the next day. After standard fixation with 4% PFA the following day, samples were subjected to two rounds of IF staining. In the first round, cells were permeabilized with 0.02% digitonin in PBS for 10 min before blocking according to the standard protocol. Following an initial incubation with the specific primary antibody, samples were incubated with species-specific secondary antibodies coupled to a cyanine-based fluorescent dye with 568 nm excitation (CF®568, Biotum). After extensive washing, cells were further permeabilized with 0.15% TritonX100 in PBS for 10 min, followed by blocking. To visualize the newly exposed epitopes, the samples were again incubated with the specific primary antibodies. This time a secondary antibody with 640 nm excitation was used (CF®640 R, Biotum).

Confocal images for the standard IF were taken on a Nikon spinning disc microscope (Yokogawa spinning disk CSU-X1) equipped with the following lasers (488 nm, 561 nm and 638 nm), a 60 × oil objective (Plan-Apo, NA 1.40 Nikon), a 40 × air objective (Plan Apo, NA 0.95), an Andor camera (AU888, 13 μm/pixel) and NIS-Elements software (Nikon).

Confocal images for the selective permeabilization experiments were taken on a Nikon spinning disc microscope (Yokogawa spinning disk CSU-W1 with 50um pinholes) equipped with the following lasers (488 nm, 561 nm and 638 nm), a 60 × oil objective (Plan-Apo LambdaD, NA 1.42 Nikon), a 40 × air objective (Plan Apo, NA 0.95), an sCMOS camera (pco.edge bi4.2, 6.5 μm/pixel) and NIS-Elements software (Nikon).

STED images were taken on a STEDYCON system (Abberior Instruments) mounted onto a Nikon TI Eclipse with a nanoZ Controller (Prior) and controlled by Micro-Manager (https://micro-manager.org/). Imaging was done with two different pulse Diode lasers (561 nm, 640 nm) for excitation paired with two single counting avalanche photodiodes (650–700 nm, 575–625 nm) for detection, a 775 nm STED laser for depletion and an optic tunable filter to modulate all laser beams for confocal imaging. Images were captured by using an 100 × 1.45 NA lambda oil objective lens with a fixed pixel size set to 20 nm for STED images.

The following primary antibodies were used in FAF2 experiment: anti-mouse-TOMM20 (1:100, Santa Cruz, sc-17764, RRID:AB 628381), anti-rat-HA (1:100, Chromotek, 7c9-100, RRID:AB_2631399), anti-rabbit-Calreticulin (1:100, Thermo, PA3-900, RRID:AB_325990), anti-rabbit-FAF2 (Proteintech, 16251-1-AP, RRID:AB_2262469) The following primary antibodies were used in FAM136A experiment: anti-mouse-TOMM20 (1:100, Santa Cruz, sc-17764, RRID:AB_628381), anti-rabbit-HA (1:100, Cayman, cay162200-1, RRID:AB_327903). The following primary antibodies were used in selective permeabilization experiment: anti-mouse-TOMM20 (1:100, Santa Cruz, sc-17764, RRID:AB_628381), anti-mouse-HA (1:100, Santa Cruz, sc-7392, RRID:AB_627809), anti-mouse-TIMM23 (1:50, BD Biosciences, 611222, RRID:AB_398754), anti-rabbit-COX4 (1:100, Cell Signaling, 4850, RRID:AB 2085424), anti-mouse-SDHA (1:50, Abcam, ab14715, RRID:AB 301433)

The following secondary antibodies were used in FAF2 and FAM136A experiment: anti-rat-AF488 (1:200, Invitrogen, A11006, RRID:AB_2534074), anti-mouse-AF594 (1:200, Thermo, A11032, RRID:AB_2534091) and anti-rabbit-AF647 (1:200, Thermo, A21244, RRID:AB_2535812). The following secondary antibodies were used in selective permeabilization experiment: anti-mouse-CF®568 (Biotum, 20105, RRID:AB 10557030), anti-rabbit-CF®568 (Biotum, 20098, RRID:AB 10557118), anti-mouse-CF®640 R (Biotum, 20177, RRID:AB 10853623), anti-rabbit-CF®640 R (Biotum, 20178, RRID:AB 10852688).

### Imaging analysis

Images were processed using FIJI analysis software (https://imagej.net/software/fiji/). Co-localization was analyzed using the Coloc2 plugin on two-color confocal images (https://imagej.net/plugins/coloc-2). Bisection threshold regression was applied to multiple manually selected ROIs per overview image with PSF 3 to determine co-localization between both acquired channels. The Pearson's R value above the threshold was reported as the co-localization metric, unless otherwise stated. For the FAF2 co-localization analysis, a dataset consisting of two conditions over four experiments with a total of 150 images was used. Co-localization was determined between the FAF2 signal and the respective organelle of interest, either mitochondria (TOMM20) or ER (Calreticulin). For the selective permeabilization assays, a dataset from 3 separate experiments with 30 images per candidate was used. Co-localization was determined between the two different permeabilization conditions per candidate.

### Computational predictions for TM regions

For the 298 first-tier interactors of the 244 LMs, we used the software TMHMM[68] to predict the transmembrane regions. Here we used the posterior probability to show the prediction confidence. 53 transmembrane proteins with known topology in 244 LMs were used to define the probability score range. Posterior probability score above 0.75 was defined as high confidence transmembrane region, 0.037–0.75 as potential transmembrane region, below 0.037 as soluble region.

### Comparison of the DSSO-cross-linking-based mitochondrial proteome to published mitochondrial proteome resources

We performed a comparative analysis on mitochondria level without considering sub-compartments. The following datasets were included:

- Our CLASP analysis of DSSO-cross-linked mitochondria, which included 748 proteins that can be confidently assigned to this organelle (albeit not always with sub-compartment information)
- The DOMs dataset from Schessner et al. (Supp. Data 1 in ref. 35), which reported 958 mitochondrial assignments
- The LOPIT-DC and HyperLOPIT datasets from Geladaki et al. (Supp. Data 1 and 2 in[36]), which assigned 464 and 548 mitochondrial proteins, respectively, in U-2 OS cells
- The MitoCoP database from Morgenstern et al. (Table S1 in ref. 34), which – after removal of duplicate entries - contains 1137 mitochondrial and mitochondrial-associated proteins
- The MitoCarta3.0 database described by Rath et al. [33] (available at https://www.broadinstitute.org/files/shared/metabolism/mitocarta/human.mitocarta3.0.html), comprising 1136 human proteins.

### Comparison of CLASP data to published proximity biotinylation datasets

We re-analyzed data from the HEK293 cell map[7], the HEK293 mitochondrial proximity interactome[29] and the IMM protein architecture map[26].

Our analysis of the Go et al. dataset was based on Supplementary Table 9 of the original paper[7], which shows cellular sub-compartment predictions for the confidently identified proximity interactors (filtered at 1% Bayesian FDR) based on two prediction algorithms – Non-negative Matrix Factorization (NMF) and Spatial Analysis of Functional Enrichment (SAFE). We filtered for proteins for which both algorithms suggested a mitochondrial localization. The number of predicted sub-compartments was determined using NMF predictions since SAFE did not provide any single sub-compartment predictions.

For the Antonicka et al. dataset, we determined the number of confidently predicted proteins by filtering their full proximity interactome (Table S4 in ref. 29) at a Bayesian false discovery rate threshold of 1% (i.e. the same cutoff as in the original publication). We then used the reported sub-compartment annotations of each BioID bait (Fig. 2B in ref. 29) to determine in which sub-compartments the reported prey proteins were detected.

For the comparison shown in Fig. 2B–D, proteins predicted to reside in a single sub-compartment were classified in "single sub-compartment" group. Proteins with ambiguous localization or predicted to reside in more than one sub-compartment were classified in "more than one sub-compartments" group (see Supplementary Data 5, column "single sub-compartment localization (Yes or No)").

For the Lee et al. dataset, we considered all transmembrane IMM proteins, for which the authors proposed or confirmed topological information (reported in Data Set S7 of[26]).

### CLASP Python tool

The developed python pipeline is divided in two parts. Users need to load an XL-MS data table and a TSV file downloaded from Swiss-prot/Uniprot with localization and topology annotations. The first part of the pipeline will automatically select LMs from proteins with unequivocal localization annotation in the Swiss-Prot database. It also offers the possibility to add and modify LMs and their annotations. The second part of the pipeline will predict protein locations and membrane protein topologies based on the provided cross-link information and the LMs defined in the first part. The resulting CLASP annotations will be automatically saved in an output table. Full instructions how to execute this tool are provided on GitHub (see Code Availability statement below).

### Reporting summary

Further information on research design is available in the Nature Portfolio Reporting Summary linked to this article.

## Data availability

All CLASP predictions and LMs are reported in Supplementary Data 5 (DSSO-cross-linked mitochondria, manual CLASP), Supplementary Data 7 (SVs), Supplementary Data 10 (DSBSO-cross-linked mitochondria), and Supplementary Data 11 (DSSO-cross-linked mitochondria, automated CLASP). The underlying cross-link identifications are reported in Supplementary Data 1–4 (DSSO-cross-linked mitochondria), Supplementary Data 6 (SVs), and Supplementary Data 9 (DSBSO-cross-linked mitochondria). The mass spectrometry raw data have been deposited to the PRIDE repository with the dataset identifiers PXD032132 and PXD046382.

For validating the mitochondrial localization of endogenous Fam136A in human cells, we used publicly available data from Human Protein Atlas (https://www.proteinatlas.org/ENSG00000035141-FAM136A/subcellular#human).

For comparing CLASP to published spatial proteomics studies, we used:

- The Go et al. dataset (Supp. Table 9 of the original paper[7])
- The Antonicka et al. dataset (Table S4 in[29])
- The Lee et al. dataset (Data Set S7 of[26])
- The DOMs dataset from Schessner et al. (Supp. Data 1 in[35])
- The LOPIT-DC and HyperLOPIT datasets from Geladaki et al. (Supp. Data 1 and 2 in[36])
- The MitoCoP database from Morgenstern et al. (Table S1 in[34])
- The MitoCarta3.0 database described by ref. 33 (available at https://www.broadinstitute.org/files/shared/metabolism/mitocarta/human.mitocarta3.0.html)

For confirming the labeling radius of DSSO-based CLASP, the following publicly available PDB structures were used:

- The mitochondrial electron transport chain complexes I, II, III and IV with PDB accession codes 5XTD, 1ZOY, 5XTE and 5Z62, respectively
- TOMM complex with PDB accession code 7CK6
- TIMM22 complex with PDB accession code 7CGP
- TIMM9-TIMM10 complex with PDB accession code 2BSK
- Succinyl-CoA ligase complex SUCLG1-SUCLG2 with PDB accession code 6G4Q
- MCAD-ETF complex with PDB accession code 1T9G
- 39S mitoribosome with PDB accession code 7OIE
- Frataxin bound iron sulfur cluster assembly complex with PDB accession code 6NZU
- Iron sulfur cluster assembly with PDB accession code 5KZ5
- Calcium uniporter homocomplex with PDB accession code 6WDN
- Transcription ignition complex with PDB accession code 6ERQ
- Mitochondrial DNA replicase with PDB accession code 4ZTU
- Trifunction protein with PDB accession code 6DV2
- FIS with PDB accession code 1PC2
- CYB5R3 with PDB accession code 1UMK
- TIMM44 with PDB accession code 2CW9
- MUT with PDB accession code 3BIC
- VDAC1 with PDB accession code 6G6U
- VDAC2 with PDB accession code 4BUM
- SLC25A13 with PDB accession code 4P5W
- COQ8A with PDB accession code 4PED
- EXOG with PDB accession code 5T5C
- CLPP with PDB accession code 6DL7
- PHB2 with PDB accession code 6IQE
- OPA1 with PDB accession code 6JTG
- TRAP1 with PDB accession code 4Z1L
- CKMT1A with PDB accession code 1QK1
- PNPase with PDB accession code 5FZ6

A source data file is provided with this manuscript. Source data are provided with this paper.

## Code availability

The Python tool for automated CLASP, a user guide, and example input and output data are provided through Github (https://github.com/theliulab/protein_location_prediction, available under a MIT license) and Zenodo (https://doi.org/10.5281/zenodo.10824759, available under a Creative Commons Attribution 4.0 International license).

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

## Acknowledgements
The work was funded by Deutsche Forschungsgemeinschaft Grant (DFG) SFB 958(Z03), DFG Project LI 3260/5-1, DFG Project LI 3260/6-1, the Leibniz-Wettbewerb (K284/2019 and P70/2018), and ERC-2020-StG (project number 949184) to F.L. as well as an FMP integrative project to M.L. and F.L.

## Author contributions
Y.Z. performed most of the experiments and data analysis. K.A. performed the confocal and STED imaging experiments. C.W., K.A. and M.R. contributed to XL-MS data generation; J.R. developed the python script for CLASP automation. D.B.L., C.W., and N.Y. assisted the experiments and data analysis. M.L. supervised the imaging analysis. Y.Z. and F.L. wrote the manuscript. F.L. developed the concept and supervised the research. All authors reviewed and edited the manuscript.

## Funding

## Competing interests
F.L. is a shareholder and advisory board member of Absea Biotechnology Ltd. and VantAI. The remaining authors declare no competing interests.
