## [Peer Review File · Nature Communications]

Cross-link assisted spatial proteomics to map sub-organelle proteomes and membrane protein topologiesEditorial Note: Parts of this Peer Review File have been redacted as indicated to remove third-party material where no permission to publish were obtained.

REVIEWER COMMENTS

Reviewer #1 (Remarks to the Author):

In the manuscript, Zhu et al. proposed a cross-link assisted spatial proteomics (CLASP) strategy to characterize multiple sub-compartments and membrane-topologies proteomes based on cross-linked distance constraints and localization markers. By CLASP, the authors analyzed the spatial proteomes of mitochondrial sub-compartments and synaptic vesicles. Especially, follow-up studies confirmed the discovery of mitochondria-associated proteins and revised previous protein localization and membrane topology data.

Overall, I think this work suitable for publication in Nature Communications. However, there are some points that the author should address:

1) The number of identified cross-links is crucial for CLASP strategy. As mentioned in the introduction, the enrichable groups of crosslinkers can greatly increase the number of identified cross-links. Why did the authors use both non-enrichable DSSO and enrichable Azide-A-DSBSO in this manuscript? And why different cross-linker concentration used in mitochondrial and synaptic vesicles analysis (0.5mM for DSSO; 2mM for Azide-A-DSBSO)?

2) The FDR estimation of inter-protein crosslinks was important for accurate determination of PPIs. 2% FDR used in this study is relatively high.

3) The CLASP labeling radius is very important to determine sub-compartments. Therefore, it might be better to move Note 1 to the main text.

4) Figure S3: The labeling radius of proximity-dependent enzymatic labeling approaches is inconsistent with many publications.

5) Figure S4B: It is suggested to provide more explanation regarding "The two cross-links between OMM and IMM proteins are depicted".

6) For CLASP newly discovered mitochondria-associated proteins, biological techniques such as Co-IP is needed to confirm the accuracy of the interactions obtained through cross-linking.

7) Figure 5: Site-specific labeling-based APEX technology has also been used for the analysis of mitochondrial membrane protein topology. Please provide the comparison with the previously published work.

8) Although Protease protection assay is a commonly used method for verifying the topology of membrane proteins, the results should be confirmed by other methods. Besides the positive control with similar localization, other confirmation experiments should be considered to ensure the accuracy and reliability of topology verification.

9) In the Discussion Section, in addition to the reliance of CLASP on LM, the coverage and accuracy of the cross-linking are crucial, which should be further clarified.

Reviewer #2 (Remarks to the Author):

Zhu et al. describe the use of cross-link assisted spatial proteomics (CLASP) to map sub-organelle proteomes and membrane protein topology using mitochondria and synaptic vesicles as model systems. Whereas several other spatial (organelle) proteomics methods previously developed, such as dynamic organellar maps, localisation of organelle proteins by isotope tagging (i.e. LOPIT) or protein correlation profiling, require subcellular fractionation; other methods relying on proximity-labelling strategies, such as BioID, TurboID or APEX can be done on intact cells in culture. Here, Zhu et al. describe CLASP, which uses covalent cross-linking coupled to liquid chromatography tandem mass spectrometry (LC-MS/MS) on purified organelles to generate distance constraints (i.e. information on protein connectivity and membrane protein topology) at a resolution (4 nm) superseding that of e.g. BioID (10 nm), APEX (20 nm) and TurboID (35 nm).

The paper is in general very well written, and provides an important development in the field of spatial proteomics and membrane protein topology using important model systems of interest to the broader scientific community. The methods and approaches used and developed are relevant, the results clearly presented and the conclusions sound. The images are of high quality. The authors additionally benchmark their findings against previously described sub-mitochondrial localizations and known protein-protein interactions to provide evidence for the robustness of their approach.

Whereas the description of the results of CLASP on isolated mitochondria provides a clear and consistent story, I fail to see what CLASP on synaptic vesicles adds to the manuscript. CLASP on mitochondria was done using the DSSO cross-linker and the resulting peptides after standard sample preparation procedures for LC-MS/MS fractionated on a strong cation exchange (SCX) column. The authors state that

they used the data for the synaptic vesicles to validate the CLASP approach on a different sample type using different workflows; for the vesicles the authors used the azide-A-DSBSO cross-linker in combination with size exclusion chromatography (SEC)-based fractionation of the peptides. Why use a different cross-linker and a different fractionation approach? And if one would use a different cross-linker to DSSO, why not use an enrichable one, such as PhoX/tBU-PhoX and omit the subsequent fractionation step applied here (SCX/SEX) and merely rely on enriching for the PhoX/tBU-PhoX cross-linked peptides? The way the manuscript is currently written, the data for the synaptic vesicles feels like an add-on, and provides little new information on the system studied – the authors themselves state that their data on the synaptic vesicles showed 100% agreement with known localization and topology of selected marker proteins.

Minor comments:

- Figure 1; i) can the authors list the top 10 proteins in the figure legend or elsewhere; ii) what information does panel E provide? It is a PPI network, but what information should a reader get from the interactome (other than that the localization markers are red and first-tier interaction blue)?
- In Figures 3A, 4A, 5 and 6A the authors depict “Predict region” in blue. What are these predict regions? Please specify in figure legend.
- Figure S2; the schematic representations are very small, and could be expanded to the width of the WB images.
- Figure S5; the text in panel A cannot be read (y- and x-axis). Consider expanding panel A to the width of the structural images (panel B) to improve legibility?

Reviewer #3 (Remarks to the Author):

Zhu et al. describe a method based on the concept that cross-links of proteins with known cellular localization could allow for the determination of subcellular localization of interacting proteins of unknown location. They name this approach CLASP (cross-link assisted spatial proteomics). To showcase the applicability of their approach, they generate two mass spectrometry datasets by analyzing mitochondria enriched from HEK293 cells and synaptic vesicles enriched from mouse brains, which were cross-linked with DSSO or Azide-A-DSBSO, respectively, and subsequently enriched/fractionated for cross-linked peptides followed by LC-MSMS and cross-link identification. In these data sets, the authors define abundant proteins yielding high numbers of cross-links with other proteins (inter-links) as markers for subcellular localization. Subsequently, they infer the subcellular localization of cross-linked proteins for which no localization existed through these identified cross-links. While no extensive bioinformatic analyses or follow up studies are performed for the synaptic vesicle data, the authors perform additional experiments for the mitochondria dataset, proposing several novel proteins to be of mitochondrial

location, for two of which they perform biological follow up experiments. Furthermore, they follow up on their prediction of protein topologies, going into detail for two more proteins.

The main claim of this paper is that the authors developed a cross-link based spatial proteomics strategy. In the eyes of this reviewer, this concept is not novel and was performed previously in several cross-linking MS papers, although it was not termed CLASP. Almost all analyses were performed for one mitochondria dataset, which are arguably the most suited organelle for such studies and whose investigation by cross-linking MS has been published. For synaptic vesicles, the authors do not show any data for possible novel interacting proteins and focus only on a handful known interactors (<2% of their dataset). Furthermore, the authors perform microscopy-based follow up experiments for only two proteins, using transient transfection experiments. For mitochondria, it seems to this reviewer that the location marker proteins were selected manually and this reviewer does not have the impression that the concept can be easily adapted by other researchers. Also, the authors do not provide any bioinformatic tools which could be used by other groups, or insights other than suggesting to select proteins with known location and many identified cross-links. This does not seem to be a major conceptual advance for the field. Therefore, this reviewer does not think that this manuscript provides a high degree of novelty and/or broad relevance, and, due to the lack of more extensive follow up experiments, also not a large number of reliable novel protein localizations which could be used by other scientists. Therefore, this article should be more suited for publication in a more specialized proteomics journal.

Major comments

The authors state that the HEK experiment was conducted in three biological replicates. How is the overlap between individual replicates for HEK cells? How many cross-links were found reproducibly? This would be an important information to be able to assess in how far CLASP enables reproducible investigation of subcellular localization. Typically, in other spatial proteomics experiments, significant enrichment is based on statistical procedures considering reproducibility and quantitative information, is something similar possible for CLASP?

The general notion in the field is that cross links define specific interactions between proteins. The authors claim here the opposite, that high abundant proteins result in high numbers of cross links with proteins which are coincidentally in their vicinity. While this is certainly plausible for highly restricted spaces with a high protein concentration such as the IMS, it is highly questionable if this will hold true for e.g. cytosolic interactions where proteins are more mobile and the co-localization by chance will be less pronounced. In line with this argument, the authors find for their synaptic vesicle preparation 5456 cross links, of which they only follow up upon 84 inter links from 28 proteins, 30% of which are from the vATPase complex whose structure at synaptic vesicles was elucidated previously by Wittig et al (ref 20 in this paper). What about all the other cross links? This reviewer wonders why no other possible interaction partners were indicated and why no follow up studies were conducted on this dataset? In

order to demonstrate that novel interaction partners could be found, such interactions should be confirmed with co-IP or fluorescence microscopy experiments. Otherwise it remains questionable in how far protein-protein interactions included in this dataset are trustworthy. Furthermore, no data are presented showing the success of synaptic vesicle enrichment e.g. by microscopy or western blot so it remains largely unclear if the material used for this experiment consisted indeed mainly of synaptic vesicles. Therefore, the general applicability to every cross-link dataset claimed by the authors, which is certainly a prerequisite to justify interest to a broad audience, does not sound realistic to this reviewer.

The authors identify 115 proteins presenting new spatial information. They do not, however, follow further up on most of these proteins. These proteins should be better evaluated with respect for published data from mitochondria, also using non proximity biotinylation approaches, e.g. from the Warscheid group or more general subcellular papers e.g. by the Lilley or Borner group. This would allow the reader to make a better use of this resource as it could be identified which of them are most likely to be true (i.e. overlaps with these datasets). Furthermore, many of these proteins are only identified based on a single cross-link, raising the question how trustworthy they are.

The authors compare their approach to BioID analyses utilizing high numbers of bait proteins (Figure 2B-D). This comparison is somehow misleading, as the authors chose publications with high numbers of bait proteins and tailored their comparison accordingly (number of IDs per experiment). While CLASP needs less different cell lines, it is still very resource demanding (the authors used 50 SCX fractions for three replicates each with 3h gradients). This restricts such analyses to material which can be obtained in high amounts (three hundred million HEK cells and 10 mouse brains were used in this study) and that few laboratories have the necessary experience and equipment, making their use very limited when compared to BioID/APEX experiments for which the necessary fusion constructs can by now be easily obtained and samples be analyzed by most core facilities with standard proteomics approaches.

The authors perform follow up analyses on two proteins, FAM136A and FAF2. This reviewer has several issues with respect to these immunofluorescence analyses. For both constructs, the authors perform transient overexpression experiments which are prone to artefacts. The authors should at least generate stable cell lines, ideally at near endogenous expression levels, or utilize antibodies which detect the endogenous proteins. Furthermore, all microscopy experiments lack a negative control which should be provided. The STED analysis (Figure 3B) lacks the resolution to unambiguously show that FAM 136 indeed localizes to the intermembrane space, an electron microscopy image would be clearly better suited. For the immunofluorescence analyses in Figure 4C, the image looks clearly overexposed and if at all, colocalization analyses can only be performed at the outer edges of the cell. Therefore, intensity based colocalization analysis in combination with Pearson's correlation will give a positive signal for sure. The parameters of the experiment should be adapted to provide better data quality and also the Mander's coefficient should be determined and provided.

The authors should provide additional follow up experiments, covering at least 2 proteins for each of the four submitochondrial localizations in order to prove the validity of their data.

The authors claim that CLASP is a universally applicable approach but do not provide any assistance for interested readers to apply it to other datasets or questions. If to be performed by another group, it comes down to manually selecting proteins of interest and manually looking at their protein-protein interactions. This is not very novel and the authors do not provide any type of framework or reference proteins for different types of subcellular localizations, e.g. by their general occurrence in cross linking experiments, which can be utilized by interested readers to aid future attempts in this direction.

Minor comments

The authors should also include other approaches than proximity biotinylation in their discussion section and data comparison.

Mitochondria where cross-linked twice while synaptic vesicles only once. What is the reason for this?

No information concerning the number of replicates is given for the synaptic vesicle preparation, this should be included.

Reviewer #4 (Remarks to the Author):

This study by Zhu et al. employs cross-linking mass spectrometry (XL-MS) to perform spatial proteomics with the goal of resolving sub-organellar protein localization and membrane protein topology, two aspects of protein localization that are still challenging to address with orthogonal approaches like proximity labeling. Building on their previous studies, the authors perform DSSO XL-MS on isolated human mitochondria, and deduced protein localization through the detection of direct crosslinks between unannotated proteins and “localization markers”, which are well characterized mitochondrial proteins. This analysis allowed for the authors to determine the sub-mitochondrial localization of 270 proteins, including 73 “new” proteins not previously assigned as mitochondrial (a major claim) and 3 mitochondria-associated cytosolic proteins. In addition, this approach provided insight to the topology of 58 mitochondrial membrane proteins, and further extended this approach to study synaptic vesicles.

The methods described in this study are technically robust and should have important utility. Additionally, the manuscript is generally well-written and the figures nicely displayed. The shortcoming of the work is that the overall findings are mostly descriptive and lack either novelty or important validation. Also, in evaluating the performance of this approach in mapping sub-mitochondrial protein localization, they compare the results to two BioID datasets, while not including more comprehensive mitochondrial proteome compendia that include similar localization information, such as MitoCarta 3.0. The authors claim that their approach reveals that FAM126A is a new mitochondrial intermembrane space protein, however this sub-organellar localization has already been documented in Mitocarta 3.0. As many of the protein localizations described in this study are documented elsewhere (e.g. MitoCarta, IMPI, MitoCoP), it is unclear what the main novelty of this approach is. The work needs to substantiate their claims of identifying an extensive new list of mitochondrial proteins and demonstrate some level of biological importance. As it stands, the cross-linking data is highly descriptive, with most observations lacking any functional follow-up experimentation or even discussion of the biological ramifications of these findings. As such, I feel the current manuscript is a strong methods paper, but probably a better fit for a more focused journal.

REVIEWER COMMENTS

General note: We have numbered all reviewer comments consecutively to make this document easier to navigate.

Reviewer #1 (Remarks to the Author):

In the manuscript, Zhu et al. proposed a cross-link assisted spatial proteomics (CLASP) strategy to characterize multiple sub-compartments and membrane-topologies proteomes based on cross-linked distance constraints and localization markers. By CLASP, the authors analyzed the spatial proteomes of mitochondrial sub-compartments and synaptic vesicles. Especially, follow-up studies confirmed the discovery of mitochondria-associated proteins and revised previous protein localization and membrane topology data. Overall, I think this work suitable for publication in Nature Communications.

We thank the reviewer for taking the time to assess our submission and for the overall positive evaluation of our work.

However, there are some points that the author should address:

1. The number of identified cross-links is crucial for CLASP strategy. As mentioned in the introduction, the enrichable groups of crosslinkers can greatly increase the number of identified cross-links. Why did the authors use both non-enrichable DSSO and enrichable Azide-A-DSBSO in this manuscript? And why different cross-linker concentration used in mitochondrial and synaptic vesicles analysis (0.5mM for DSSO; 2mM for Azide-A-DSBSO)?

In principle, CLASP can be applied to any proteome-wide XLMS dataset. We felt that this versatility is best demonstrated by applying CLASP to different XLMS datasets (cell lines vs primary tissue) generated via very distinct workflows (DSSO vs DSBSO and SCX fractionation vs affinity enrichment + SEC fractionation). Since DSSO and DSBSO have distinct physicochemical properties, individually optimized crosslinking conditions are necessary to obtain the best possible results and are the reason for the differences in crosslinker concentrations. We have found that adding DSSO in two aliquots of 0.5 mM yields the best crosslinking results for human cell and organelle samples, and have used this procedure for many years (see for example Wang et al. EMBO Rep 2020, 10.15252/embr.201948833 / Ruwolt et al., Anal Chem 2022, 10.1021/acs.analchem.1c04812). Similarly, we have optimized the DSBSO crosslinking protocol in-house in the context of a different, still unpublished project (see below).

Figure R 1. Example for the optimization of DSBSO crosslinking conditions

We fully agree with the reviewer that enrichable crosslinkers can increase the depth of XLMS experiments and that this is beneficial for CLASP. Therefore, we have also acquired a XLMS dataset of DSBSO-crosslinked mitochondria (Table R 1). Of note, for this experiment we deliberately used the same total crosslinker concentration as for the DSSO experiment (1 mM). As shown below, the DSBSO-based CLASP predictions capture the majority of overall spatial annotations (431 out of 542, Figure R 2) and unambiguous localization predictions (163 out of 270, Figure R 3) of the DSSO dataset. Importantly, when unambiguous CLASP predictions were possible based on both datasets, predictions agreed in nearly all cases (160 out of 163). This further supports the robustness of the CLASP approach.

Datasets	Identified cross-links (on residue-pair level; 2%FDR with separate filtering of inter-links and intra-links)	LMs candidates selected	Agreement of inter-LM cross-links with reported LM localization	LMs selected (after conflict filtration)	Number of first-tier interactors of LMs	Fraction of unambiguous localization predictions for first-tier interactors
DSSO	13971 (7721 inter-protein)	258	99.7% (4490/4502)	244	298	91% (ambiguous: 28/298)
DSBSO	26650 (15224 inter-protein)	253	99.7% (7188/7209)	235	420	93% (ambiguous: 28/420)
Overlap				224	186	

Table R 1. Comparison of the DSSO- and DSBSO-based CLASP results for intact mitochondria.

Figure R 2. Overlap of spatially annotated proteins in DSSO-based CLASP and DSBSO-based CLASP. This plot includes internally consistent LMs (i.e. without conflicting cross-links) and their first-tier interactors.

A CLASP predicted proteins (without ambiguous) in DSSO and DSBSO datasets

B Distribution of CLASP predictive consistency in DSSO and DSBSO datasets

Figure R 3. Consistency between DSSO-based and DSBSO-based CLASP predictions. (A) Overlap of proteins, for which CLASP predictions could be made using the DSSO and DSBSO mitochondrial XLMS data. (B) Number of proteins with consistent and inconsistent prediction results in DSSO-based and DSBSO-based CLASP.

Action taken: We include the CLASP analysis of the DSBSO dataset in the revised manuscript (see new Results sub-section “CLASP with an enrichable cross-linker validates and deepens spatial insights into mitochondria”). Furthermore, we clarify in the revised manuscript that different crosslinking conditions are a consequence of the different physicochemical properties of the crosslinkers and the samples being crosslinked (see Methods sub-section “Cross-linking of synaptic vesicles”).

2. The FDR estimation of inter-protein crosslinks was important for accurate determination of PPIs. 2% FDR used in this study is relatively high.

The best way to control FDR is a complex topic that is still controversially discussed in the XLMS community. For instance, FDR can be controlled on different levels (cross-link spectrum matches, unique residue pairs, or PPIs). False positives increase proportionally when moving to higher levels (see <https://doi.org/10.1021/acs.analchem.6b03745>). For example, a 1% FDR cutoff on the spectrum level will result in residue-pair and PPI FDRs clearly above 1%. Furthermore, the optimal cutoff often depends may depend on whether other filtering criteria were applied to ensure high annotation confidence.

To reach a good compromise between identification sensitivity and specificity, we filtered our data at 2% FDR on the level of unique residue pairs. In our view, this is well in line with previous XLMS studies from different research groups on mitochondria and synaptic vesicles which imposed a 1-5% FDR on spectrum level (e.g. Linden et al., MCP 2020, 10.1074/mcp.RA120.002028 / Wittig et al., Nat Commun 2021, 10.1038/s41467-021-21102-w / Sun et al., Anal Chem 2022, 10.1021/acs.analchem.2c02116) or a 5% FDR on residue pair level (Ryl et al., JPR 2020, 10.1021/acs.jproteome.9b00541). Importantly, FDR filtering was done separately for inter-links and intra-links to avoid an inflation of false-positive inter-links.

It also worth noting that we not only imposed an FDR but also applied additional filtering criteria. First, we removed protein clusters that are disconnected from the main interaction network because these may lead to spurious localization annotations. Second, we limited the CLASP predictions to proteins that are directly cross-linked to localization markers (blue dots in Fig 1E), omitting all indirectly connected proteins (grey dots in Fig 1E). As a result, our CLASP predictions are based on a densely connected protein network and the vast majority of them is supported by multiple cross-links (Figure R 9) and confirmed by CLASP predictions based on an independent mitochondrial XLMS dataset generated using DSBSO cross-linker (Figure R 2). This increases their confidence. We hope the reviewer will agree that this strategy is suitable in the context of our CLASP analysis. Finally, we make all our raw data publicly available, so everyone will be able to re-analyze them with the FDR control approach of their choice.

3. The CLASP labeling radius is very important to determine sub-compartments. Therefore, it might be better to move Note 1 to the main text.

We fully agree with the reviewer and have modified the manuscript accordingly.

Action taken: Moved Supplementary Note 1 to the main text (see new Results subsection “Cross-linking provides a well-defined maximum labeling radius”).

4. Figure S3: The labeling radius of proximity-dependent enzymatic labeling approaches is inconsistent with many publications.

We are unsure which specific publications the reviewer is referring to, but we agree that there the precise labelling radii of proximity biotinylation approaches are not entirely clear. The numbers in this supplementary figure are derived from the references cited in the figure legend:

- *BioID: Kim et al. (10.1073/pnas.1406459111) used the known arrangement of Y-Nup subunits in the nuclear pore complex to infer a labelling radius of 10 nm*
- *APEX: Martell et al. (10.1038/nbt.2375) used APEX to visualize individual intermediate filaments with ~20 nm repeat spacing suggesting a labelling radius in this range. More recently, Oakley et al. (10.1073/pnas.2203027119) determined by super-resolution microscopy a much larger labelling radius of 269 ± 41 nm.*
- *Contact-ID: Kwak et al. (10.1073/pnas.1916584117) estimated a 10-20 nm labelling radius without supporting evidence*
- *TurboID: May et al. (10.3390/cells9051070) used the same approach as Kim et al. for BioID and found that the labelling radius of TurboID is substantially larger (>35 nm)*

If the reviewer is aware of more accurate labelling radius estimates, we will be happy to include them in our paper.

5. Figure S4B: It is suggested to provide more explanation regarding "The two cross-links between OMM and IMM proteins are depicted".

We apologize that that the description was not sufficiently clear and hope that the revised legend explains things better.

Action taken: We rephrased the legend (now Supplementary Fig 2B).

6. For CLASP newly discovered mitochondria-associated proteins, biological techniques such as Co-IP is needed to confirm the accuracy of the interactions obtained through cross-linking.

It is important to note that CLASP is designed to provide information on protein proximity, whereas IP/AP identifies physical protein interactions based on affinity. As such, we would not expect both methods to provide the same results. This supported by data from an independent study focused on herpes simplex virus type 1 (HSV-1), where we systematically compared results XLMS and AP-MS (Bogdanow et al., biorxiv 2023, <https://doi.org/10.1101/2023.12.03.569778>) . As shown in the figure below, overlap between both methods is only around 50% even for modest AP-MS enrichment cutoffs of 1-2 log₂ fold change. Therefore, we do not feel that IP/AP would be the optimal validation technique in this case.

Figure R 4. Percentage overlap between protein connections confirmed by XLMS and AP-MS at different AP-MS enrichment cutoffs. The XLMS data are limited to tier-1 interactors of the proteins selected as AP-MS baits. Data adopted from <https://doi.org/10.1101/2023.12.03.569778>. [REDACTED]

Furthermore, the mitochondrial proteome is composed of a large number of membrane proteins. One of the emphases in our study is that XLMS covers very well the interactions of membrane proteins, which are very challenging to study by AP/IP experiments. This advantage of XLMS allows us to predict membrane protein topologies by CLASP and many of our follow-up experiments focus membrane proteins, which are not amenable to AP/IP. Instead of using AP/IP, we had validated one of the three identified mitochondria-associated proteins – FAF2 (also called UBXD8) – with protease protection assays and confocal imaging (Fig 4), providing evidence that FAF2 localizes to ER and mitochondria. We published these data in May 2022 on biorxiv (<https://doi.org/10.1101/2022.05.05.490733>). A few months later, this finding was independently confirmed in another publication (Zheng et al., EMBO Rep 2022, 10.15252/embr.202254859). We feel that these findings strongly support the accuracy of our identifications and predictions.

Action taken: We have expanded the discussion around FAF2 to include the recent independent work on this protein (see Results sub-section “CLASP discovers mitochondria-associated proteins”).

7. Figure 5: Site-specific labeling-based APEX technology has also been used for the analysis of mitochondrial membrane protein topology. Please provide the comparison with the previously published work.

We agree that this is an important comparison. It was already included in Supplementary Fig 3A of the original submission and discussed in the Results section. Briefly, 17 protein topologies were determined in both the APEX study (Lee et al., JACS 2017,

10.1021/jacs.6b10418) and our paper, and 14 of them (82%) are consistent. Furthermore, 43 APEX annotations are not predicted by CLASP. Most of these proteins (24) have well established topologies, which is why CLASP considers them as localization markers and not as predictions. Conversely, CLASP gave topology information for 25 IMM and 16 OMM proteins not provided in the APEX dataset (see also Results sub-section “CLASP characterizes the topologies of membrane proteins” and the display items referenced therein).

Importantly, the APEX study was specifically designed to map membrane protein topologies, whereas CLASP provided topological information and sub-compartment localization data in parallel.

8. Although Protease protection assay is a commonly used method for verifying the topology of membrane proteins, the results should be confirmed by other methods. Besides the positive control with similar localization, other confirmation experiments should be considered to ensure the accuracy and reliability of topology verification.

We agree that, in principle, multiple lines of validation are desirable to when establishing a new method. However, hardly any complementary method offers the resolution required to determine sub-compartment localization or membrane topology (not even super-resolution microscopy as also pointed out by Reviewer#3 – comment 32). Therefore, protease protection assays are the gold standard in mitochondrial spatial biology and, for example, have also been used to confirm sub-compartment localization predictions in the BioID-based mitochondrial proximity interactome (Antonicka et al. Cell Metab 2020, 10.1016/j.cmet.2020.07.017).

Nonetheless, we have performed additional validation experiments:

- I) As already described in our original submission, we confirmed the IMS location of FAM136A by alkaline carbonate extraction.
- II) We followed up on the CLASP localization prediction for NIPSNAP2 and the CLASP membrane topology prediction for TMEM126A with selective permeabilization experiments.

Selective permeabilization assays rely on treatment with digitonin (permeabilizes only OMM) or triton (permeabilizes OMM and IMM), followed by confocal imaging to test under which detergent conditions the proteins become accessible for detection antibodies. OMM and IMS proteins become readily detectable upon digitonin treatment, whereas matrix proteins can only be detected in the presence of triton.

The imaging results are displayed below (Figure R 5). The detection pattern of C-terminally tagged TMEM126A resembles those of IMS and OMM markers, confirming our CLASP topology prediction that the TMEM126A termini face the IMS. NIPSNAP2, for which previous studies gave controversial results suggesting either OMM (Shanmughapriya et al., Mol Cell 2015, 10.1016/j.molcel.2015.08.009) or matrix (Princely Abudu et al., Dev Cell 2019, 10.1016/j.devcel.2019.03.013) localization, only becomes accessible upon triton treatment. The same is true for matrix marker SDHA and COX4, which has a matrix-facing antibody-binding region. This confirms the CLASP prediction that NIPSNAP2 is a matrix protein, which we additionally validated by a protease protection assay.

Figure R 5. Selective permeabilization experiments to determine the sub-compartment localizations of TMEM126A and NIPSNAP2.

(A) Confocal dual-color images after immunofluorescence staining. Each channel shows respective staining of the candidate in dependence of the permeability reagent using highly cross-absorbed secondary antibodies. Shown are representative images from one out of three biological replicates with antibody and plasmid expressing controls. Image brightness and contrast are adjusted to the same Look-Up Table scale in all images. OMM protein TOMM20, IMM proteins TIMM23 and FAM162A, and matrix protein SDHA are used as markers for each mitochondrial sub-compartment. Scale bar: 10 μ m.

(B) Pearson correlation coefficients (PCC) for signal colocalization between digitonin and triton treatment conditions for manually selected regions of interest within the overview images from (A). Correlation is high for OMM proteins and proteins with antibody binding regions in the IMS since they become accessible under both conditions. Correlation is decreased for proteins with antibody-binding regions in the matrix since they only become available in the presence of triton. NIPSNAP2-HA PCC values resemble matrix those of markers SDHA; TMEM126A PCC values resemble those of IMM markers FAM162A and TIMM23, confirming their CLASP predictions. Shown are the mean with the SEM and the individual data points from three biological replicates, distinguishable by the shape of the points (11 images per experiment, with multiple cells each per overview).

Figure R 6 Localizing NIPSNAP2 to the mitochondrial matrix. (A) Cross-link map of NIPSNAP2 and its interacting proteins based on DSSO and DSBSO cross-linking. LMs are shown in green; NIPSNAP2 is shown in purple. The CLASP annotation of NIPSNAP2 is supported by direct connections to 2 Matrix LMs and 3 IMM LMs. Proteins that are not LMs but support the M localization of NIPSNAP2 are shown in grey. (B) Protease protection assay combined with WB to analyze the localization of NIPSNAP2-HA in HEK293T cells. OMM protein TOMM70, IMM protein TIMM23 and matrix protein SDHA are used as markers for each mitochondrial sub-compartment.

Action taken: We have added the selective permeabilization data and the NIPSNAP2 protease protection assay as Supplementary Figures 6 and 7.

9. In the Discussion Section, in addition to the reliance of CLASP on LM, the coverage and accuracy of the cross-linking are crucial, which should be further clarified.

We agree with the reviewer that these are important considerations for any CLASP analysis. We tried to address this aspect in the Discussion section of the previous version as follows: "It is important to note that the accuracy of CLASP annotations critically depends on the comprehensiveness and interconnectivity of the XL-based protein network. [...] Therefore, CLASP can readily provide detailed localization annotations for any purifiable organelles but would likely be less powerful when applied to limited sample amounts (e.g. primary samples) or intact cells, since proteome coverage of most in-cell XLMS workflows is currently still limited."

We acknowledge that we only referred to the accuracy of CLASP but not the accuracy of the XLMS dataset.

Action taken: We revised the above sentence to "It is important to note that the power of CLASP critically depends on the comprehensiveness, accuracy, and interconnectivity of the XL-based protein network."

Reviewer #2 (Remarks to the Author):

Zhu et al. describe the use of cross-link assisted spatial proteomics (CLASP) to map sub-organelle proteomes and membrane protein topology using mitochondria and synaptic vesicles as model systems. Whereas several other spatial (organelle) proteomics methods previously developed, such as dynamic organellar maps, localisation of organelle proteins by isotope tagging (i.e. LOPIT) or protein correlation profiling, require subcellular fractionation; other methods relying on proximity-labelling strategies, such as BioID, TurboID or APEX can be done on intact cells in culture. Here, Zhu et al. describe CLASP, which uses covalent cross-linking coupled to liquid chromatography tandem mass spectrometry (LC-MS/MS) on purified organelles to generate distance constraints (i.e. information on protein connectivity and membrane protein topology) at a resolution (4 nm) superseding that of e.g. BioID (10 nm), APEX (20 nm) and TurboID (35 nm). The paper is in general very well written, and provides an important development in the field of spatial proteomics and membrane protein topology using important model systems of interest to the broader scientific community. The methods and approaches used and developed are relevant, the results clearly presented and the conclusions sound. The images are of high quality. The authors additionally benchmark their findings against previously described sub-mitochondrial localizations and known protein-protein interactions to provide evidence for the robustness of their approach.

We are happy to hear that the reviewer appreciates our work and thank them for their comments.

10. Whereas the description of the results of CLASP on isolated mitochondria provides a clear and consistent story, I fail to see what CLASP on synaptic vesicles adds to the manuscript. CLASP on mitochondria was done using the DSSO cross-linker and the resulting peptides after standard sample preparation procedures for LC-MS/MS fractionated on a strong cation exchange (SCX) column. The authors state that they used the data for the synaptic vesicles to validate the CLASP approach on a different sample type using different workflows; for the vesicles the authors used the azide-A-DSBSO cross-linker in combination with size exclusion chromatography (SEC)-based fractionation of the peptides. Why use a different cross-linker and a different fractionation approach? And if one would use a different cross-linker to DSSO, why not use an enrichable one, such as PhoX/tBU-PhoX and omit the subsequent fractionation step applied here (SCX/SEX) and merely rely on enriching for the PhoX/tBU-PhoX cross-linked peptides? The way the manuscript is currently written, the data for the synaptic vesicles feels like an add-on, and provides little new information on the system studied – the authors themselves state that their data on the synaptic vesicles showed 100% agreement with known localization and topology of selected marker proteins.

We agree that the synaptic vesicle (SV) data were not well connected to the rest of the paper. As correctly stated by this reviewer, the purpose of the SV analysis was to demonstrate that CLASP is not restricted to a single biological system or a specific XLMS workflow. The reviewer mentions that we should have used an enrichable cross-linker and this is precisely what we did: DSBSO-linked peptides were enriched using dibenzocyclooctyne (DBCO)-coupled sepharose beads (see Methods section). SEC fractionation of the enriched sample was performed subsequently to further increase the depth of our analysis. We could have performed a similar experiment with tBU-PhoX (not with PhoX because it is not membrane-

permeable). However, since (tBU)-PhoX is shorter and more rigid, it yields more intra-protein than inter-protein links (see also our original paper on tBU-PhoX, <https://doi.org/10.1002%2Fanie.202113937>), which we think may not be the best choice for CLASP, where only inter-protein links provide spatial information of the proteins connected to localization markers.

The reviewer is also correct that the CLASP analysis of SVs did not yield new biological information. However, our primary goal with this analysis was validation and not biological discovery. We felt that SVs are well suited for this purpose because – while their overall proteome is highly dynamic and not fully understood – their constitutive proteome is small and well characterized. This property allowed us to further confirm the internal consistency and thus validity of our localization marker concept, which is CLASP's central underlying premise. All that said, we now realize that placing this validation experiment at the end of the Results section was misleading and illogical. This has been amended in the revised manuscript.

Action taken: We integrated the sub-section on SVs into the first Results sub-section (“CLASP enables detailed protein localization mapping based on routine XL-MS”) that focuses on validating the CLASP concept. The corresponding Fig 7 is now shown as Supplementary Fig 5. We also emphasize more strongly that this is a validation experiment.

Minor comments:

11. - Figure 1; i) can the authors list the top 10 proteins in the figure legend or elsewhere; ii) what information does panel E provide? It is a PPI network, but what is the information should a reader get from the interactome (other than that the localization markers are red and first-tier interaction blue)?

(i) The top 50 proteins were already highlighted in Supplementary Data 5. We now also highlight the top 10 proteins in this file and mention them in the figure legend.

(ii) Panel E is meant to visualize the size and interconnectivity of the XLMS-derived mitochondrial protein network (as mentioned in the discussion, CLASP critically depends on comprehensive datasets with many inter-protein connections. It also shows how much of the network is covered by the localization markers (red) and their first-tier interactors (blue, i.e. the proteins for which we predict localizations). We now clarify this in the figure legend.

Action taken: Revised Supplementary Data 5 and the legends of Figures 1B and 1E as described above.

12. - In Figures 3A, 4A, 5 and 6A the authors depict “Predict region” in blue. What are these predict regions? Please specify in figure legend.

We apologize for this confusion. In all figures except for Figure 5, “Predict region” referred to the protein/protein region, for which a CLASP prediction was made. In Figure 5, “Predict region” referred to transmembrane regions predicted by TMHMM2.0

Action taken: We changed the coloring and labelling in Figure 5 to make it distinguishable from the “Predict region” label in the other figures. In all figures that show the “Predict region” label, we clarify its meaning in the legends.

13. - Figure S2; the schematic representations are very small, and could be expanded to the width of the WB images.

Action taken: Fixed as requested (now Supplementary Fig 8).

14. - Figure S5; the text in panel A cannot be read (y- and x-axis). Consider expanding panel A to the width of the structural images (panel B) to improve legibility?

Action taken: Fixed as requested (now Supplementary Fig 3).

Reviewer #3 (Remarks to the Author):

Zhu et al. describe a method based on the concept that cross-links of proteins with known cellular localization could allow for the determination of subcellular localization of interacting proteins of unknown location. They name this approach CLASP (cross-link assisted spatial proteomics). To showcase the applicability of their approach, they generate two mass spectrometry datasets by analyzing mitochondria enriched from HEK293 cells and synaptic vesicles enriched from mouse brains, which were cross-linked with DSSO or Azide-A-DSBSO, respectively, and subsequently enriched/fractionated for cross-linked peptides followed by LC-MSMS and cross-link identification. In these data sets, the authors define abundant proteins yielding high numbers of cross-links with other proteins (inter-links) as markers for subcellular localization. Subsequently, they infer the subcellular localization of cross-linked proteins for which no localization existed through these identified cross-links. While no extensive bioinformatic analyses or follow up studies are performed for the synaptic vesicle data, the authors perform additional experiments for the mitochondria dataset, proposing several novel proteins to be of mitochondrial location, for two of which they perform biological follow up experiments. Furthermore, they follow up on their prediction of protein topologies, going into detail for two more proteins.

We thank this reviewer for their detailed comments on our work.

15. The main claim of this paper is that the authors developed a cross-link based spatial proteomics strategy. In the eyes of this reviewer, this concept is not novel and was performed previously in several cross-linking MS papers, although it was not termed CLASP.

We feel this assessment doesn't do full justice to the CLASP concept. The idea for CLASP arose from the realization that crosslinkers can be used as molecular rulers to measure the distances between proteins and thus can infer spatial information of proteins in relation to known markers. This principle is similar to available proximity labelling approaches such as BioID and APEX, however CLASP has the advantages that 1) it is distance-based rather than diffusion-based and therefore more accurate, and 2) CLASP affords a much shorter labelling radius (e.g. 1 nm for DSSO) and thus higher spatial resolution than BioID (10 nm according to Kim et al., PNAS 014, 10.1073/pnas.1406459111) and APEX (269 ± 41 nm according to Oakley et al., PNAS 2022, 10.1073/pnas.2203027119).

In fact, the distances that can be bridged by commonly used crosslinkers such as DSSO are even smaller than the radius of the highest resolution proximity labeling approach developed to date (μ Map, labelling radius ca. 4 nm, see Geri et al., Science 2020, 10.1126/science.aay4106). While μ Map can only be used outside of cells, we and others have established that several cross-linkers can penetrate different biomembranes without compromising membrane integrity. Based on this, we hypothesized that XLMS has the potential to serve as a high-resolution spatial proteomics approach inside biological compartments. To make this possible, we developed CLASP as a data analysis framework based on the newly introduced concept of localization markers (LMs). This enabled the first systematic demonstration that XLMS data can reveal (1) protein localizations at the sub-compartment level and (2) topological information for membrane proteins. To the best of our knowledge, no previous XLMS study has provided this kind of information. As the reviewer mentions in their comment 25 below, so far XLMS data have been used almost exclusively to

characterize specific protein interactions; the only protein localization data previously derived from XLMS was on compartment level – i.e. proteins were tentatively assigned to a specific organelle because they were found cross-linked to other proteins that are known to reside in this organelle.

16. Almost all analyses were performed for one mitochondria dataset, which are arguably the most suited organelle for such studies and whose investigation by cross-linking MS has been published.

We chose mitochondria as our main model system for three reasons:

- 1) *They consist of two soluble subcompartments (IMS and matrix) and two membranes (IMM and OMM), which allowed us to validate the ability of CLASP to provide both spatial information with sub-compartment resolution and membrane protein topology information in parallel.*
- 2) *Purified mitochondria have been extensively characterized using APEX and BioID, i.e. the only other spatial proteomics approaches that can yield sub-compartment spatial information and insights into membrane protein topology. As a result, several high-quality mitochondrial APEX and BioID datasets are available that we could use for direct benchmarking of CLASP.*
- 3) *Several BioID and APEX-based methods have been initially validated using mitochondria (e.g. Kwak et al., PNAS 2020, 10.1073/pnas.1916584117 / Lee et al., JACS 2017, 10.1021/jacs.6b10418 / Rhee et al., Science 2013, 10.1126/science.1230593). We felt that using the same system would be the fairest way to validate CLASP.*

In addition, we analysed synaptic vesicles to demonstrate that the essential prerequisite CLASP – the detection and internal consistency of cross-linked LMs – is fulfilled in samples other than mitochondria.

Action taken: We revised the final paragraph of the introduction to explain in more detail why we chose mitochondria as our main model system.

17. For synaptic vesicles, the authors do not show any data for possible novel interacting proteins and focus only on a handful known interactors (<2% of their dataset).

We agree that the synaptic vesicle (SV) data were not well explained and presented in the original submission. For more information on the purpose of this analysis, please refer to our response to Reviewer#2 (comment 10). Briefly, we included the SV analysis to demonstrate that the validity of the LM concept, which represents the core of CLASP, is not restricted to mitochondria and one specific XLMS workflow. Therefore, we focused the analysis on constitutive SV resident proteins that can serve as LMs. We should have stated more clearly that the aim of this experiment is method validation and not biological discovery, and it was misleading to present these data at the end of the Results section. This has been amended in the revised manuscript.

Action taken: We integrated the sub-section on SVs into the first Results sub-section (“CLASP enables detailed protein localization mapping based on routine XL-MS”) that focuses on validating the CLASP concept. The corresponding Fig 7 is now shown as Supplementary Fig 5). We also emphasize more strongly that this is a validation experiment.

18. Furthermore, the authors perform microscopy-based follow up experiments for only two proteins, using transient transfection experiments.

The goal of this study was to determine protein sub-compartment localizations and membrane topologies. Microscopy is not ideally to validate these findings because – as this reviewer correctly points out below (comment 32) – even super-resolution microscopy cannot unambiguously confirm localization to sub-compartments like the IMS. Similar problems arose when we attempted immunogold staining-EM, which is understandable considering that the size of most gold particles (ca. 5-25 nm) is in the same range as the diameters of the smaller sub-compartments (IMM/OMM: 7 nm, see 10.1006/jsbi.1997.3885 / IMS: 14-28 nm, see 10.1016/S0968-0004(00)01609-1).

Therefore, we mostly relied on biochemical validation by protease protection assays – the gold standard for confirming sub-mitochondrial localization – which was done for 3 OMM proteins (FAF2, CYB5R3 and CYB5B), 1 IMS protein (FAM136A, additionally validated by alkaline carbonate extraction), 1 IMM protein (TMEM126A) in the original submission. In the revision, we validated 5 additional protein localizations using this assay (see response to comment 34). Nonetheless, we included an additional imaging-based follow-up experiment: We combined confocal imaging with selective permeabilization by digitonin and triton to further validate the membrane topology of TMEM126A and confirm the matrix localization of NIPSNAP2 (see our response to Reviewer#1’s comment 8). Furthermore, we performed another XLMS experiment in mitochondria using an enrichable crosslinker (DSBSO) and CLASP analysis of these data confirmed many of our original predictions.

Action taken: We include a CLASP analysis of an XLMS dataset of DSBSO-crosslinked mitochondria (see new Results sub-section “CLASP with an enrichable cross-linker validates and deepens spatial insights into mitochondria”), a comparison of these data to our original CLASP predictions (Supplementary Figures 11 and 12), and a selective permeabilization assay supporting the CLASP predictions for TMEM126A and NIPSNAP2 (Supplementary Figure 7).

19. For mitochondria, it seems to this reviewer that the location marker proteins were selected manually and this reviewer does not have the impression that the concept can be easily adapted by other researchers. Also, the authors do not provide any bioinformatic tools which could be used by other groups, or insights other than suggesting to select proteins with known location and many identified cross-links. This does not seem to be a major conceptual advance for the field.

As discussed in our manuscript, CLASP LMs fulfil a similar role as baits in APEX/BioID approaches. We selected the LMs manually, because manual selection of bait proteins based on existing biological knowledge is common practice in APEX/BioID. For their mitochondrial

BioID dataset, Antonicka et al. manually selected 100 baits from MitoCarta2.0 based on two criteria: (1) they should cover all mitochondrial sub-compartments, (2) they should be “likely to illuminate new aspects of mitochondrial biology” (Antonicka et al. *Cell Metab* 2020, 10.1016/j.cmet.2020.07.017). Similarly, bait selection for the BioID human cell map was fully manual (Go et al., *Nature* 2020, 10.1038/s41586-021-03592-2).

As shown in Figure R 7 below, most of the LMs we could validate are highly abundant proteins. This makes sense because highly abundant proteins (1) tend to be cross-linked to more interactors (see Figure R 7) and (2) are more likely to be biologically well characterized, which increases the chance that their sub-compartment localization is known. Therefore, the cross-links of highly abundant proteins are overall more likely to provide rich spatial information. We consider it a strength of CLASP that it can take advantage of highly abundant well-characterized proteins to generate information on proteins that are less well understood.

Figure R 7. Abundance of cross-linked proteins plotted against their number of detected interactors for the DSSO mitochondria dataset. LMs are shown in red, other proteins are shown in blue. Abundance is based on the average iBAQ value from three biological replicates.

That being said, we find the reviewer’s suggestion for a computational tool to support CLASP analyses very valuable. Therefore, we developed a Python pipeline that will automatically select LMs using Uniprot information (all reviewed human protein entries as of July 2023) and subsequently perform a CLASP analysis using any XLMS dataset as input. The tool along with a full description is available through Github (https://github.com/theliulab/protein_location_prediction).

We applied this new Python tool to our DSSO-crosslinked mitochondria dataset and compared the resulting CLASP predictions based on our original approach with manual LM selection:

LM picking	LMs candidates selected	Agreement of inter-LM cross-links with reported LM localization	LMs selected (after conflict filtration)	Number of first-tier interactors of LMs	Coverage of XL-based network by LMs + first-tier interactors	Fraction of unambiguous localization predictions for first-tier interactors
Manual (original approach – see Supplementary Data 5)	258	99.7% (4490/4502)	244	298	72%	91% (ambiguous: 28/298)
Automated (script)	88	98.8% (1016/1028)	72	271	45%	95% (ambiguous: 14/271)

Table R 2. Comparison of CLASP annotations using manual or script-based LM selection.

Reassuringly, the automatically selected LM candidates show very high internal consistency (98.8% of cross-links agreeing with their reported location) and yield almost only unambiguous localization predictions (95%). The vast majority of the automatically obtained LMs and first-tier interactors have also been found using our original manual approach. For 98% (292 proteins) of the overlapping identifications, automated and manual CLASP give the same annotation of sub-compartment localization and membrane topology. Furthermore, performing automated CLASP with our manually selected LMs results in predictions that are 100% identical to the ones originally reported. These results demonstrate that automated LM selection and CLASP predictions work robustly.

Figure R 8. Overlap between CLASP LMs and first-tier interactors based on the original manual approach and the script-based automated approach.

Table R 2 also shows that the Python tool selects fewer LMs and thus yields fewer CLASP predictions. This is expected since the Python tool only considers Uniprot data, which are unlikely to contain all available biological information. Therefore, users still have the option to add manually selected LMs. For instance, following the reasoning described above, users could take abundance as a metric to identify additional potential LMs, and confirm their suitability through a literature and /or PDB search. Generally speaking, a higher number of LMs will increase the cross-links that support the CLASP predictions (see figure below) and thereby increase the prediction accuracy.

Figure R 9. Distribution of spatially annotated proteins that are supported by different numbers of LM cross-links.

Action taken: We provide the CLASP Python tool and a user guide through Github (https://github.com/theliulab/protein_location_prediction). A brief description of the tool is also included in the manuscript (see Results sub-section “A Python workflow for automated CLASP predictions”). Figure R 8 has been added as Supplementary Fig 14.

20. Therefore, this reviewer does not think that this manuscript provides a high degree of novelty and/or broad relevance, and, due to the lack of more extensive follow up experiments, also not a large number of reliable novel protein localizations which could be used by other scientists. Therefore, this article should be more suited for publication in a more specialized proteomics journal.

We respectfully disagree with this assessment. We have addressed the novelty concerns in our response to this reviewer’s comment 15.

We are unsure why the reviewer questions the reliability of our localization predictions. We had already provided several lines of evidence that the CLASP results are robust:

- *Cross-links among our LMs showed near-perfect agreement with the established localization/topology of these proteins (99.7% for DSSO-crosslinked mitochondria, 100% for DSBSO-crosslinked SVs)*
- *79% of our spatial annotations are supported by more than one cross-link (see Figure R 9, red bars).*
- *80% of overlapping protein sub-compartment annotations by CLASP and BioID (Antonicka et al.) are consistent*
- *82% of overlapping membrane topology annotations by CLASP and APEX (Lee et al., JACS 2017, 10.1021/jacs.6b10418) are consistent*
- *5 novel CLASP predictions had already been validated by complementary biochemical or imaging experiments.*

In the revised manuscript, we have added more data confirming our CLASP predictions:

- *CLASP analysis of DSBSO-crosslinked mitochondria confirming the majority of our previous predictions derived from DSSO crosslinking (see response to Reviewer#1’s comment 1).*

- *Selective permeabilization experiment to further validate the CLASP predictions for TMEM126A and NIPSNAP2 (see response to Reviewer#1's 8th comment)*
- *Complementary biochemical validation of NIPSNAP2 and 4 additional CLASP predictions (see response to Reviewer#3's comment 34)*

We hope that, collectively, these results will resolve the reviewer's concerns over the reliability of CLASP.

Action taken: We include the CLASP analysis of an XLMS dataset of DSBSO-crosslinked mitochondria (see new Results sub-section "CLASP with an enrichable cross-linker validates and deepens spatial insights into mitochondria"), a comparison of these data to our original CLASP predictions (Supplementary Figures 11 and 12), protease protection assays and alkaline carbonate extraction data for 4 additional proteins localized by CLASP (Figure 8), selective permeabilization assays supporting the CLASP predictions for TMEM126A and NIPSNAP2 (Supplementary Figure 7), and a protease protection assay further validating the NIPSNAP2 data (Supplementary Figure 6).

Major comments

21. The authors state that the HEK experiment was conducted in three biological replicates. How is the overlap between individual replicates for HEK cells? How many cross-links were found reproducibly? This would be an important information to be able to assess in how far CLASP enables reproducible investigation of subcellular localization.

The overlap between individual replicates was already presented in the original submission (Supplementary Fig 1) and is similar to other recent proteome-wide XLMS studies in eukaryotic systems (cited in the legend of Supplementary Fig 1).

To further demonstrate the reproducibility of CLASP, we have conducted another mitochondria XLMS experiment using DSBSO as a crosslinker and done a CLASP analysis of these data. Because DSBSO allows enrichment of the cross-links, the CLASP predictions based on this dataset are more comprehensive than our original DSSO-based predictions. As shown in the response to Reviewer#1's first comment (Table R 1, Figure R 2, Figure R 3), DSBSO-based CLASP captured the vast majority of the original DSSO-based CLASP results and the both datasets are highly consistent in their overlapping localization predictions.

Action taken: We include the CLASP analysis of an XLMS dataset of DSBSO-crosslinked mitochondria (see new Results sub-section "CLASP with an enrichable cross-linker validates and deepens spatial insights into mitochondria") and a comparison of these data to our original CLASP predictions (Supplementary Figures 11 and 12)

22. Typically, in other spatial proteomics experiments, significant enrichment is based on statistical procedures considering reproducibility and quantitative information, is something similar possible for CLASP?

All existing spatial proteomics methods derive spatial information from quantitative proteomics data and therefore must rely on statistical methods to determine which proteins are

significantly enriched in one location vs another. By contrast, CLASP derives spatial predictions directly from qualitative XLMS data (i.e. cross-link identifications that pass the FDR cutoff and other filtering criteria). For this reason, CLASP does not require quantitative information. That said, the CLASP concept can, in principle, be applied to any comprehensive proteome-wide XLMS dataset from an intact biological system. Workflows for quantitative XLMS are being actively developed (e.g. Ruwolt et al., *Anal Chem* 2022, 10.1021/acs.analchem.1c04812 / Chavez et al., *Anal Chem* 2020, 10.1021/acs.analchem.0c03128) and quantitative proteome-wide XLMS can be used to study changes of protein localizations (e.g., translocation of proteins upon stimulations), which however is not the subject of this manuscript.

Action taken: We clarify in the third paragraph of the Discussion section that CLASP as presented in this study exclusively relies on qualitative XLMS data.

23. The general notion in the field is that cross links define specific interactions between proteins. The authors claim here the opposite, that high abundant proteins result in high numbers of cross links with proteins which are coincidentally in their vicinity. While this is certainly plausible for highly restricted spaces with a high protein concentration such as the IMS, it is highly questionable if this will hold true for e.g. cytosolic interactions where proteins are more mobile and the co-localization by chance will be less pronounced.

*In our view, crosslinking has always been recognized as a proximity labelling approach in contrast to affinity-based approach like AP/IP. This is manifested in the general agreement in the XLMS community that a cross-link can occur whenever the applied cross-linker is long enough to bridge the distance between two amino acid residues with appropriate functional groups. It is generally accepted that proximity-based labelling (and spatial proteomics data in general) can provide functionally relevant insights (for a comprehensive overview, see for example Christopher et al., *Nat Rev Methods Primers* 2021, 10.1038/s43586-021-00029-y). Proximity-based crosslinks are no different – they are specific and reflect biological reality. Therefore, crosslinks can provide the same type of information as other proximity labelling approaches such as BioID and APEX, which are widely recognized for their ability to characterize both protein interactions and localizations (see for example Gingras et al., *Curr Opin Chem Biol* 2019, 10.1016/j.cbpa.2018.10.017; Chen et al., *Wiley Interdiscip Rev Dev Biol* 2017, 10.1002/wdev.272). While crosslink-based information on protein interactions has been well explored, CLASP is the first approach to leverage crosslink-based information on protein localization. As discussed in response to this reviewer’s comment 15, CLASP offers the advantages of 1) a clear-cut labelling radius based on a defined distance instead of a diffusion-based labelling radius and 2) a substantially higher spatial resolution than BioID and APEX. In our manuscript, we provide evidence for both the specific labelling radius (see Results sub-section “Cross-linking provides a well-defined maximum labeling radius”) and the high spatial resolution (low fraction of ambiguous sub-compartment assignments – Figure 2B-D, capability to map membrane topologies – Figures 5 and 6).*

The reviewer may be correct that CLASP may not work in a dilute environment of equally distributed proteins, but there is no spatial information to be gleaned from such an environment, meaning that no spatial proteomics approach would be able to produce meaningful localization data. Furthermore, we think the reviewer might underestimate the

protein concentration in the cytosol. Based on our calculation of protein concentrations in different compartments below, cytoplasm has even a slightly higher protein concentration compared to IMS.

Our estimation of the relative protein concentrations in the cytoplasm and mitochondrial sub-compartments is based on the following data:

- 1) Mitochondria protein copy numbers from MitoCoP (Morgenstern et al., Cell Metabol 2021, 10.1016/j.cmet.2021.11.001). Since total copy numbers in MitoCoP are higher than the sum of the sub-compartment-specific copy numbers, we extrapolated the latter to match the total number, using the percentage of copy numbers assigned to each sub-compartment.
- 2) Cytoplasmic protein copy numbers from OpenCell map (Cho et al., Science 2022, 10.1126/science.abi6983)
- 3) Relative mitochondria and cytoplasm volumes from Table 12-1 in Alberts B, Johnson A, Lewis J, et al. Molecular Biology of the Cell. 4th edition. New York: Garland Science; 2002. The Compartmentalization of Cells. Available from: <https://www.ncbi.nlm.nih.gov/books/NBK26907/>
- 4) Relative volumes of sub-compartments within mitochondria from <https://doi.org/10.1101/2022.08.16.500715> (Discussion section) which presents a correction of the values reported in Garcia et al., Sci Rep 2019, <https://doi.org/10.1101/2022.08.16.500715>

	MitoCoP sc-specific copy numbers reported	MitoCoP sc-specific copy numbers %	copy numbers extrapolated to total MitoCoP proteome	relative volume	relative concentration (copy number/vol)
OMM	2.10E+07	0.072524	2.45E+07	0.02	1.23E+09
IMS	6.34E+06	0.021915	7.41E+06	0.022	3.37E+08
IMM	1.25E+08	0.431668	1.46E+08	0.052	2.81E+09
Matrix	1.37E+08	0.473893	1.60E+08	0.106	1.51E+09
Total Mito	2.89E+08	1	3.38E+08	0.2	1.69E+09
			OpenCell copy numbers reported	relative volume	relative concentration (copy number/vol)
Cytoplasm			2.92E+08	0.54	5.41E+08

Table R 3. Protein copy number and relative concentration estimates for mitochondrial sub-compartments and the cytosol.

Based on these numbers, we believe if CLASP works in IMS, it should, in principle, also be applicable in the cytoplasmic protein environment.

24. In line with this argument, the authors find for their synaptic vesicle preparation 5456 cross links, of which they only follow up upon 84 inter links from 28 proteins, 30% of which are from the vATPase complex whose structure at synaptic vesicles was elucidated previously by Wittig et al (ref 20 in this paper). What about all the other cross links? This reviewer wonders why no other possible interaction partners were indicated and why no follow up studies were conducted on this dataset? In order to demonstrate that novel interaction partners could be found, such interactions should be confirmed with co-IP or fluorescence microscopy experiments. Otherwise it remains questionable in how far protein-protein interactions included in this dataset are trustworthy.

The sole reason for including the SV dataset was to demonstrate that the LM concept remains valid for a different XLMS workflow and in a different subcellular compartment (see also our responses to comments 10 and 17). We would like to emphasize that the SV experiment was not designed for biological discovery. For example, different from Wittig et al., we omitted the final ultracentrifugation step, which would have yielded a more concentrated SV preparation that may have yielded deeper biological insights. We agree that this was not sufficiently explained in the original submission and that placing the SV analysis at the end of the Results section was misleading.

The full SV XLMS dataset is included in Supplementary Data 6 and the raw data are available on PRIDE (see Data Availability statement). The reviewer will find that the vast majority of the remaining crosslinks involve cytosolic proteins. This is expected because the synaptic proteome is known to be highly variable and to include many "SV-visitor" proteins (see Taoufiq et al., PNAS 2020, 10.1073/pnas.2011870117). At the same time, we cannot exclude that some unrelated cellular proteins have been co-purified.

Since the goal of the SV experiment was to validate the LM concept, we restricted our analysis to well-characterized proteins that can serve as LMs. We found our cross-links to be in full agreement with the known localization/topology of these LMs, which proves that our data are trustworthy.

Action taken: We integrated the sub-section on SVs into the first Results sub-section ("CLASP enables detailed protein localization mapping based on routine XL-MS") that focuses on validating the CLASP concept. The corresponding Fig 7 is now shown as Supplementary Fig 5. We also emphasize more strongly that this is a validation experiment.

25. Furthermore, no data are presented showing the success of synaptic vesicle enrichment e.g. by microscopy or western blot so it remains largely unclear if the material used for this experiment consisted indeed mainly of synaptic vesicles. Therefore, the general applicability to every cross-link dataset claimed by the authors, which is certainly a prerequisite to justify interest to a broad audience, does not sound realistic to this reviewer.

We apologize for this omission. We have followed the standard protocol for SV purification (Takamori et al., Cell 2006, 10.1016/j.cell.2006.10.030 / Huttner et al., J Cell Bio 1983, 10.1083/jcb.96.5.1374). We have followed the same protocol to prepare another SV sample and analyzed it by negative-stain EM micrograph (shown below, along with a micrograph of the SV preparation from Wittig et al. before their final sample concentration step). The quality of both preparations is similar.

Figure R 10. Negative stain EM micrographs of the SV preparation following the protocol in this study (left) and by Wittig et al. (taken from Fig S1A in <https://doi.org/10.1038/s41467-021-21102-w>) [REDACTED]
We have analyzed the same SV preparation in an independent, collaborative study by cryoET. We are including a representative tomogram below for the reviewer's information because it provides clearer evidence for the structural integrity of the SVs.

Figure R 11. Cryo-electron tomogram of the SV preparation. Left: A slice through a cryo-ET volume of a hippocampal cell cultured on a grid with a putative presynapse containing individual SVs. Right: A slice through a cryo-ET volume of isolated SVs showing synaptic vesicles. Scale bars = 100 nm. Orange arrows indicate V-ATPases on the SV surface (adopted from Kravcenko, Ruwolt et al., manuscript under review). [REDACTED]

Action taken: We include the negative stain EM micrograph of our SV preparation as Supplementary Fig 16. Furthermore, we had mistakenly cited the wrong reference for SV preparation in the Methods sub-section “Synaptic vesicles purification” and have amended this now.

26. The authors identify 115 proteins presenting new spatial information. They do not, however, follow further up on most of these proteins. These proteins should be better evaluated with respect for published data from mitochondria, also using non proximity biotinylation approaches, e.g. from the Warscheid group or more general subcellular papers e.g. by the Lilley or Borner group. This would allow the reader to make a better use of this resource as it could be identified which of them are most likely to be true (i.e. overlaps with these datasets).

The added value of CLASP lies in its ability to provide insights into sub-compartment localization and membrane topology. The only other approaches that provide such insights are proximity biotinylation methods, which is why we focused our comparative analyses on BioID and APEX. The MitoCoP database from the Warscheid lab as well as DOM (Borner lab) and LOPIT (Lilley lab) only assign mitochondrial proteins without further sub-division (see next paragraph) and thus do report the level of spatial information that CLASP is providing.

To still address the reviewer’s question, we performed a comparative analysis on mitochondria level without considering sub-compartments. Our CLASP analysis of DSSO-crosslinked mitochondria included 748 proteins, which can be confidently assigned to this organelle (albeit not always with sub-compartment information). This compares to mitochondrial assignments in other spatial proteomics datasets as follows:

- *The most recent DOM paper by the Borner lab assigned 958 mitochondrial proteins (see Supplementary Data 1 of Schlessner et al., Nat Commun 2023, 10.1038/s41467-023-41000-7)*
- *The Lilley lab assigned 464 mitochondrial proteins by LOPIT-DC and 548 mitochondrial proteins by HyperLOPIT in U-2 OS cells (see Supplementary Data 1 and 2, Geladaki et al., Nat Commun 2019, 10.1038/s41467-018-08191-w)*
- *The Warscheid lab combined different proteomics approaches and different cell lines to define a comprehensive mitochondrial proteome. Their spatial proteomics approach in itself yielded 978 mitochondria-linked proteins. Combining subtractive and spatial proteomics, importomics, and literature/database curation, they define a MitoCoP database “of 1,274 mitochondrial and mitochondria-associated proteins including isoforms.” (Morgenstern et al., Cell Metabol 2021, 10.1016/j.cmet.2021.11.001). After removing duplicates, we arrived at 1137 mitochondrial and mitochondrial-associated proteins.*

A comparison to these resources as well as the Mitocarta 3.0 database is shown below.

Figure R 12. Venn diagram of identified mitochondrial proteins from all datasets mentioned above

The Venn diagram shows that a core of 286 common proteins (19% of all proteins in all datasets). This represents 38% of our DSSO dataset. Furthermore, 73% of the mitochondrial proteins found by DSSO cross-linking have been identified in at least one published dataset.

Action taken: We included these analyses as Supplementary Fig 4.

27. Furthermore, many of these proteins are only identified based on a single cross-link, raising the question how trustworthy they are.

We apologize for not describing more clearly how well our CLASP predictions are supported. As shown in our response to comment 19 (Figure R 9, red bars), 79% of the spatial annotations made based on our DSSO data are supported by more than one cross-link (counted as unique residue-pairs). In addition, the vast majority of the DSSO-based CLASP results could be confirmed in an independent CLASP analysis of DSBSO-crosslinked mitochondria (see our response to Reviewer#1's comment 1).

Action taken: We include the CLASP analysis of an XLMS dataset of DSBSO-crosslinked mitochondria (see new Results sub-section “CLASP with an enrichable cross-linker validates and deepens spatial insights into mitochondria”) and a comparison of these data to our original CLASP predictions (Supplementary Figures 11 and 12). In addition, we have indicated the number of unique residue-pair cross-links supporting each CLASP prediction in Supplementary Data 5 (DSSO dataset) and 10 (DSBSO dataset).

28. The authors compare their approach to BioID analyses utilizing high numbers of bait proteins (Figure 2B-D). This comparison is somehow misleading, as the authors chose publications with high numbers of bait proteins and tailored their comparison accordingly (number of IDs per experiment).

As mentioned in our previous responses, the aim of CLASP is to provide spatial information on the sub-compartment and membrane topology level. Therefore, we could only compare our results to BioID and APEX dataset that offer this degree of spatial granularity. Looking at published proximity biotinylation studies, it appears to us that a high number of baits are simply necessary to unambiguously assign multiple sub-compartments in parallel. This becomes clear when comparing the two BioID datasets we analysed: Go et al. (Nature 2021, 10.1038/s41586-021-03592-2) used 12 mitochondrial baits but were only able to distinguish three categories (matrix, IMM/IMS and OMM/peroxisome). Antonicka et al. (Cell Metab. 2020, 10.1016/j.cmet.2020.07.017), on the other hand, used 100 baits and were able to separate all mitochondrial sub-compartments. Earlier mitochondrial proximity biotinylation studies that used fewer baits (e.g. Rhee et al., Science 2013, 10.1126/science.1230593 and Hung et al., Mol Cell 2014, 10.1016/j.molcel.2014.06.003) did not attempt to study multiple sub-compartments and can thus not be used for comparison.

29. While CLASP needs less different cell lines, it is still very resource demanding (the authors used 50 SCX fractions for three replicates each with 3h gradients). This restricts such analyses to material which can be obtained in high amounts (three hundred million HEK cells and 10 mouse brains were used in this study) and that few laboratories have the necessary experience and equipment, making their use very limited when compared to BioID/APEX experiments for which the necessary fusion constructs can by now be easily obtained and samples be analyzed by most core facilities with standard proteomics approaches.

*We agree that our XLMS analysis still required a substantial amount of LC-MS run time (3 replicates * 50 fractions * 180 min = 450 hours). In comparison, Antonicka et al. (i.e. the only other analysis that was able to resolve all mitochondrial sub-compartments) analysed 2-8 replicates of 100 baits with a 125 min LC gradient, thus their analysis required a minimum of 417 hours run time (assuming only 2 replicates for all baits). However, different from CLASP, the Antonicka et al. study did not provide insights into membrane protein topology. It is also important to note that runtime will vary greatly with experimental design. For example, our new mitochondrial DSBSO dataset achieved greater depth than the DSSO dataset based on LC-MS analysis of only 42 fractions with a 180-min gradient (=126 hours), thus greatly reducing the required resources. In addition, we feel it should not be discounted that CLASP saves a lot of time by making all predictions from a single type biological sample that does not need to*

be genetically engineered in any way. As such, we find these LC-MS run time comparisons not particularly informative and would prefer to not include them in the paper.

We also agree that CLASP – just like any other XLMS-based approach – currently requires relatively high amounts of input materials. We had already mention in the Discussion that APEX/BioID has undisputed advantages for in vivo applications and that CLASP is of limited utility when sample amounts are limited, and we have further emphasized this point in the revised manuscript.

We do not think that equipment is a major limitation here. The XLMS experiments for CLASP require widely established enrichment techniques (SCX, SEC and/or affinity enrichment), standard LC-MS equipment, and any available cross-link identification software, several of which are now included in widely used proteomics software (e.g. MS Annika and XlinkX in Proteome Discoverer, MaxLynx in MaxQuant). In addition, we now provide a freely available Python pipeline to automated LM selection and CLASP localization predictions (see our response to this reviewer’s comment 19).

While we agree that experience will increase the success rate of CLASP studies, we believe that this is the case for any newly developed method. It does not appear to us that APEX and BioID could be readily implemented into the standard workflows of proteomics core facilities when these methods were first reported ca. 10 years ago. Similarly, the reason why many BioID/APEX fusion constructs are now commercially available is many researchers adopted these methods after their utility was demonstrated in initial proof-of-concept studies. We present here the first proof-of-concept for using cross-linking as a spatial proteomics tool and we are confident that this approach will be of interest to many other scientists inside and outside the XLMS field. In our view, one aspect that substantially lowers the barrier for others to try out CLASP is that it only requires standard, non-engineered cells as starting material.

Action taken: We emphasize more strongly that CLASP may not be applicable when sample amounts are limited (see final paragraph of the Discussion). We also provide a CLASP Python tool and a user guide through Github (https://github.com/theliulab/protein_location_prediction). A brief description of the tool is also included in the manuscript (see Results sub-section “A Python workflow for automated CLASP predictions”). A comparison of manual and tool-assisted CLASP is shown in Supplementary Fig 14.

30. The authors perform follow up analyses on two proteins, FAM136A and FAF2. This reviewer has several issues with respect to these immunofluorescence analyses. For both constructs, the authors perform transient overexpression experiments which are prone to artefacts. The authors should at least generate stable cell lines, ideally at near endogenous expression levels, or utilize antibodies which detect the endogenous proteins.

Thank you for this suggestion. The mitochondrial localization of endogenous FAM136A has previously been confirmed in multiple human cell lines (see <https://www.proteinatlas.org/ENSG00000035141-FAM136A/subcellular#human>) but the sub-compartment localization of FAM136A was unknown and could predicted by CLASP. Since confocal and super-resolution microscopy cannot confidently resolve different sub-

compartments (as also mentioned by this reviewer in comment 32), we used a protease protection assay and alkaline carbonate extraction to confirm FAM136A's IMS localization. We have confirmed the mitochondrial localization of endogenous FAF2 using antibody stainings (see below).

Figure R 13. Antibody staining of endogenous FAF2, confirming it's colocalization with mitochondrial marker TOMM20. Figure R 14. Left panel: Confocal dual-color microscopy overview (scale bar is 10 μ m). Middle panel: cropped images of HeLa cells stained for endogenous FAF2 (magenta) and TOMM20 (cyan) using immunofluorescence (scale bar is 5 μ m). Right panel: Confocal single-color microscopy overview images from (A). The scale bar is 10 μ m.

Action taken: The above-mentioned reference to public imaging data of endogenous Fam136A has been added to the manuscript (sub-section “CLASP reveals biologically relevant sub-compartment localizations”); the IF images of the FAF2 endogenous antibody staining has been added to Fig. 4D.

31. Furthermore, all microscopy experiments lack a negative control which should be provided.

We apologize for this omission and have fixed this.

Action taken: We have added negative controls to Figures 3B (confocal imaging of FAM136-HA) and 4C (confocal imaging of FAF2-HA).

32. The STED analysis (Figure 3B) lacks the resolution to unambiguously show that FAM 136 indeed localizes to the intermembrane space, an electron microscopy image would be clearly better suited.

We agree that the STED images are not fully conclusive. Therefore, we confirmed the IMS localization of FAM136A by two complementary biochemical approaches – alkaline carbonate extraction and protease protection assays – which are the generally accepted methods in mitochondrial biology to confirm protein sub-compartment localizations (e.g. also used by Antonicka et al. to confirm localization of EXD2 isoforms – see Figures 6 and S6 in 10.1016/j.cmet.2020.07.017).

We attempted immunogold-labeling EM experiments for several proteins but unfortunately these experiments were inconclusive, conceivably because the size of most gold particles (ca. 5-25 nm) is in the same range as the diameters of the smaller sub-compartments (IMM/OMM: 7 nm, see 10.1006/jsbi.1997.3885 / IMS: 14-28 nm, see 10.1016/S0968-0004(00)01609-1).

33. For the immunofluorescence analyses in Figure 4C, the image looks clearly overexposed and if at all, colocalization analyses can only be performed at the outer edges of the cell. Therefore, intensity based colocalization analysis in combination with Pearson's correlation will give a positive signal for sure. The parameters of the experiment should be adapted to provide better data quality and also the Mander's coefficient should be determined and provided.

We understand the reviewer's concern. However, the brightness of the image is a result of increased contrast and not overexposure. We have adjusted the brightness and contrast settings, and included a negative control (see below).

Figure R 15. Confocal imaging of FAF2-HA. Negative control is shown on the left, Colocalization analysis with TOMM20 and Calreticulin is shown on the right.

Colocalization analysis was only done at the cell periphery and we apologize for not mentioning this explicitly before. Pearson's (PCC) and Mander's (MCC) correlation analyses are shown below. Since the use of Mander's coefficient is not universally accepted in the field and PCC is seen as superior (e.g. Adler and Parmryd, Cytometry A 2010 and 2021, 10.1002/cyto.a.20896 and 10.1002/cyto.a.24336), we decided to only include the PCC plots in the revised manuscript. We hope the reviewers and editors will agree with this choice.

Figure R 16. FAF2 colocalization analysis using Pearson's (left) and Mander's (right) correlation. Each symbol type represents a datapoint from a separate biological replicate (4 replicates with 10–15 cells each). Colored symbols indicate means of the individual replicates. Colored lines indicate the mean across all replicates. Data have been acquired outside of the peri-nuclear region.

Action taken: We have adjusted the contrast setting, included an untransfected negative control for FAF2-HA staining, and re-done the Pearson's correlation analyses (Figure 4C, E).

34. The authors should provide additional follow up experiments, covering at least 2 proteins for each of the four submitochondrial localizations in order to prove the validity of their data.

As mentioned above, we had already performed follow-up experiments for 3 OMM proteins (FAF2, CYB5R3 and CYB5B), 1 IMS protein (FAM136A, additionally validated by alkaline carbonate extraction), and 1 IMM protein (TMEM126A). We have now additionally validated

- the matrix localization of NIPSNAP2 by selective permeabilization and protease protection assays (see response to Reviewer#1's comment 8)
- the membrane topology of TMEM126A by selective permeabilization experiments (see response to Reviewer#1's comment 8)
- the localization of 3 IMS and 1 matrix protein, which were predicted from our new DSBSO mitochondrial dataset, using protease protection and alkaline carbonate extraction assays (see below)

Figure R 17. DSBSO-based CLASP reveals CNP as a new matrix protein, and MIF, MARCKS and PFN1 as new IMS proteins.

(A) Cross-link map of CNP, MIF, MARCKS, PFN1 and their interacting proteins. LMs are shown in green; The predicted proteins are shown in purple. Proteins that are not LMs but support the predicted localization are shown in grey.

(B-E, left panel) Alkaline carbonate extraction of mitochondria isolated from HEK293T cells overexpressing CNP-HA, MIF-HA, MARCKS-HA, PFN1-HA separately. The OMM protein TOMM70, IMM protein TIMM23 and IMM associated protein TIMM44 are used as markers for each sub-compartment. T, total mitochondrial extraction; S, supernatant; P, pellet of mitochondrial membrane.

(B-E, right panel) Protease protection assay combined with WB to analyze the localization of CNP-HA, MIF-HA, MARCKS-HA, PFN1-HA in HEK293T cells separately. OMM protein TOMM70, IMM protein TIMM23 and matrix protein MRPS18B are used as markers for each mitochondrial sub-compartment.

Action taken: We added protease protection assays and alkaline carbonate extraction data for 4 additional proteins localized by CLASP (Fig. 7). The validation for NIPSNAP 2 and TMEM126A are included in Supplementary Figures 6 and 7.

35. The authors claim that CLASP is a universally applicable approach but do not provide any assistance for interested readers to apply it to other datasets or questions. If to be performed by another group, it comes down to manually selecting proteins of interest and manually looking at their protein-protein interactions. This is not very novel and the authors do not provide any type of framework or reference proteins for different types of subcellular localizations, e.g. by their general occurrence in cross linking experiments, which can be utilized by interested readers to aid future attempts in this direction.

Please refer to our response to this reviewer's comment 19. Briefly, we understand the reviewer's concern and now provide a Python pipeline that automates the LM selection and CLASP prediction process. That said, we still would recommend future CLASP users to optimize their prediction results manually fine tune their LM selection, taking into account the specificities of their biological system and their XLMS dataset. In this regard, protein abundance and connectivity are useful indicators for selecting LMs, as we also show in response to comment 19.

Action taken: We provide a CLASP Python tool and a user guide through Github (https://github.com/theliulab/protein_location_prediction). A brief description of the tool is also included in the manuscript (see Results sub-section "A Python workflow for automated CLASP predictions"). A comparison of manual and tool-assisted CLASP is shown in Supplementary Fig 14.

Minor comments

36. The authors should also include other approaches than proximity biotinylation in their discussion section and data comparison.

Please refer to our response to this reviewer's comment 26. Proximity biotinylation methods are the only approaches that can provide localization information on the sub-compartment and membrane topology level, which is the central purpose of CLASP. Since other spatial proteomics approaches do not provide such information (as already mentioned in the introduction), we don't think a more detailed discussion is warranted. To address still address

this request, we include a comparison of overall mitochondrial proteome coverage by CLASP and other spatial proteomics resources.

Action taken: We included these comparisons as Supplementary Fig 4.

37. Mitochondria were cross-linked twice while synaptic vesicles only once. What is the reason for this?

Since DSSO and DSBSO have distinct physicochemical properties, individually optimized crosslinking conditions are necessary to obtain the best possible results and are the reason for the differences in crosslinker concentrations (see our response to Reviewer#1's comment 1 for more details). In our hands, DSSO crosslinking is most effective when adding the crosslinker in two aliquots of 0.5 mM during the incubation time, resulting in a final DSSO concentration of 1 mM. By contrast, SVs were crosslinked by adding DSBSO in a single 2mM aliquot.

Action taken: We clarify in the Methods section what we mean by “cross-linked twice” (see Methods sub-section “Cross-linking of mitochondria”).

38. No information concerning the number of replicates is given for the synaptic vesicle preparation, this should be included.

Added as requested, we apologize for this oversight. Both validation experiments (DSBSO crosslinking of SVs and DSBSO crosslinking of mitochondria) were performed once.

Action taken: We have added this information to the Methods sub-sections “Cross-linking of mitochondria” and “Cross-linking of synaptic vesicles”.

Reviewer #4 (Remarks to the Author):

This study by Zhu et al. employs cross-linking mass spectrometry (XL-MS) to perform spatial proteomics with the goal of resolving sub-organellar protein localization and membrane protein topology, two aspects of protein localization that are still challenging to address with orthogonal approaches like proximity labeling. Building on their previous studies, the authors perform DSSO XL-MS on isolated human mitochondria, and deduced protein localization through the detection of direct crosslinks between unannotated proteins and “localization markers”, which are well characterized mitochondrial proteins. This analysis allowed for the authors to determine the sub-mitochondrial localization of 270 proteins, including 73 “new” proteins not previously assigned as mitochondrial (a major claim) and 3 mitochondria-associated cytosolic proteins. In addition, this approach provided insight to the topology of 58 mitochondrial membrane proteins, and further extended this approach to study synaptic vesicles.

The methods described in this study are technically robust and should have important utility. Additionally, the manuscript is generally well-written and the figures nicely displayed.

We thank the reviewer for appreciating the utility and robustness of our CLASP method

39. The shortcoming of the work is that the overall findings are mostly descriptive and lack either novelty or important validation.

This work is largely focused on developing a new spatial proteomics method and providing a resource for mitochondrial biology. As such, while we agree that the work is to some extent descriptive, we think that this is inherent to such method- and resource-centric studies.

We believe the main value of CLASP lies in its ability to give insights into membrane protein topologies as well as protein localization at the sub-compartment level and in its potential as a discovery tool. This utility has already been demonstrated for FAF2 (UBXD8), which CLASP predicted as a new mitochondria-associated proteins (originally published in May 2022 on biorxiv (<https://doi.org/10.1101/2022.05.05.490733>) and which has since been confirmed in an independent publication (Zheng et al., EMBO Rep 2022, 10.15252/embr.202254859).

Regarding your concerns on novelty and validation, please also refer to our responses to Reviewer#3's comments 15 and 20.

Action taken: For further validation, we include a CLASP analysis of an XLMS dataset of DSBSO-crosslinked mitochondria (see new Results sub-section “CLASP with an enrichable cross-linker validates and deepens spatial insights into mitochondria”), a comparison of these data to our original CLASP predictions (Supplementary Figures 11 and 12) as well as protease protection and alkaline carbonate extraction assays for 4 additional proteins localized by CLASP (Figure 8), and a selective permeabilization assay supporting the CLASP predictions for TMEM126A and NIPSNAP2 (Supplementary Figure 7). We have also expanded the discussion around FAF2 to include the recent independent work on this protein (see Results sub-section “CLASP discovers mitochondria-associated proteins”).

40. Also, in evaluating the performance of this approach in mapping sub-mitochondrial protein localization, they compare the results to two BioID datasets, while not including more comprehensive mitochondrial proteome compendia that include similar localization information, such as MitoCarta 3.0.

We had two reasons to focus on comparing CLASP to two BioID and one APEX dataset (the latter for membrane topology predictions):

First, we believe the best way to benchmark the power of a new method is to compare it to what other single methods can achieve in a similar biological system. While MitoCarta is a highly valuable database, it cumulates localization assignments from numerous experimental approaches and biological systems, which is why we felt it is not directly comparable to CLASP.

Second, we found that many MitoCarta entries lack sub-compartment annotation or provide sub-compartment annotations that are not fully compliant with the literature (see also our response to the next comment), which hampers a fair side-by-side comparison.

That said, we still find the reviewer's suggestion valuable. To implement it, we have opted to perform a comparison to MitoCarta3.0, MitoCoP and other spatial proteomics resources on the organelle level (i.e. disregarding CLASP's sub-compartment and membrane topology predictions. The results are shown in our response to Reviewer#3's comment 26 and have been included in the revised manuscript.

Action taken: We include these comparisons as Supplementary Fig 4.

41. The authors claim that their approach reveals that FAM126A is a new mitochondrial intermembrane space protein, however this sub-organellar localization has already been documented in Mitocarta 3.0.

The reviewer is correct that FAM136A is annotated as an IMS protein in Mitocarta 3.0. However, the only evidence for this annotation comes from one of the earliest APEX studies that sought to define the IMS proteome (Hung et al., Mol Cell 2014, 10.1016/j.molcel.2014.06.003). This study did not take into account other mitochondrial sub-compartments and the main evidence for its IMS assignments was protein enrichment compared to a cytosolic APEX control. As such, it does not provide direct evidence for enrichment of FAM136A in the IMS vs. other mitochondrial sub-compartments. To our knowledge, the only other evidence for FAM136A's mitochondrial localization comes from confocal microscopy (<https://www.proteinatlas.org/ENSG00000035141-FAM136A/subcellular#human>) and Requena et al., Hum Mol Genet 2015, 10.1093/hmg/ddu524), which did not reveal the sub-compartment localization.

We would argue that our CLASP analysis, which covered all mitochondrial sub-compartments, together with our validation experiments (protease protection assay, alkaline carbonate extraction, and STED microscopy) provides the first definitive evidence for FAM136A's IMS localization.

Action taken: We clarify in the manuscript that FAM136A is annotated as an IMS protein in MitoCarta 3.0, but that unequivocal experimental proof is still missing (see Results sub-section “CLASP reveals biologically relevant sub-compartment localizations”).

42. As many of the protein localizations described in this study are documented elsewhere (e.g. MitoCarta, IMPI, MitoCoP), it is unclear what the main novelty of this approach is.

As explained in our response to this reviewer's comment 39 and Reviewer#3's comments 15, the main novelty of CLASP lies in its ability to allocate proteins to mitochondrial sub-compartments and, in parallel, provide membrane protein topology information. There is no other method that can generate such insight from a single experiment. At the same time, we agree that many of the detected proteins have been assigned to mitochondria previously (albeit without sub-compartment annotation), which in our view validates that our initial XLMS experiment captured a representative mitochondrial proteome.

43. The work needs to substantiate their claims of identifying an extensive new list of mitochondrial proteins and demonstrate some level of biological importance.

We assume the reviewer is referring to the claim made in the introduction that 73 new mitochondrial proteins have been discovered. We apologize for not explaining this in more detail. Of all proteins for which we could make CLASP predictions (mitochondrial sub-compartment and/or membrane topology) based on our DSSO dataset, 76 have not been assigned to mitochondria according to Uniprot, MitoCarta3.0, and MitoCoP. Of these 76 proteins, 3 are predicted to be mitochondria-associated and 73 are predicted to be mitochondrial. However, we agree that this is an imperfect measure, especially because there may still be literature evidence that localizes some of these proteins to mitochondria but has not been incorporated into the above-mentioned databases yet. Moreover, these numbers will change again if we consider the additional discoveries from our new CLASP analysis of DSBSO-cross-linked mitochondria. Having reconsidered this issue, we feel that reporting numbers without context in the introduction distracts from the main purpose of CLASP, which is assigning protein sub-compartment localizations and membrane topologies and not necessarily expanding organellar proteomes (traditional proteomics approaches are more for this purpose since they are more sensitive than XLMS).

As mentioned above, the biological relevance of our work arises from CLASP's value as a discovery tool that can inform biological follow-up studies. An example for this is FAF2 (UBXD8), for which we discovered a dual localization to ER and mitochondria. Meanwhile, this has been independently confirmed (Zheng et al., EMBO Rep 2022, 10.15252/embr.202254859) and FAF2 has been shown to regulate ER-mitochondria contact sites (Ganji et al., Nat Commun 2023, 10.1038/s41467-023-36298-2).

Action taken: We have revised the final paragraph of the Introduction to remove all number related claims and focus instead on summarizing the CLASP concept, the analyzed XLMS datasets, and the orthogonal validation approaches. In addition, we have expanded the discussion around FAF2 to include the recent independent work on this protein (see Results sub-section “CLASP discovers mitochondria-associated proteins”).

44. As it stands, the cross-linking data is highly descriptive, with most observations lacking any functional follow-up experimentation or even discussion of the biological ramifications of these findings. As such, I feel the current manuscript is a strong methods paper, but probably a better fit for a more focused journal.

We thank this reviewer for reiterating that this is a strong methods paper, because this is precisely what we were aiming for. We have expanded our follow-up experiments (see our response to this reviewer’s comment 39) and have included additional discussion of how CLASP-based discoveries can be a starting point for new functional discoveries, using FAF2 as an example (see our response to this reviewer’s comment 43). That said, we agree that the main achievements of this study lie in spatial proteomics method development and resource generation. Considering that many papers with this scope have been published in Nature Communications over the past years (e.g. Schlessner et al., Nat Commun 2023, 10.1038/s41467-023-41000-7 / Zhai et al., Nat Commun 2022, 10.1038/s41467-022-32689-z / Ke et al., Nat Commun 2021, 10.1038/s41467-020-20367-x / Martinez-Val et al., Nat Commun 2021, 10.1038/s41467-021-27398-y / Geladaki et al., Nat Commun 2019, 10.1038/s41467-018-08191-w), we believe that this journal would be an excellent home for our work but of course this is ultimately for the editor to decide.

REVIEWER COMMENTS

Reviewer #1 (Remarks to the Author):

The author's revisions have greatly enhanced the quality of the article, yet there is a key issue that must be addressed before publication, as also highlighted by the comments of several other reviewers. The CLASP labeling radius is very important for determining sub-compartments, and within this paper, especially in the descriptions in the main text and the results figures, the emphasis on the argumentation should be increased, including:

1)The author refers to different literature reports regarding the labeling radius of proximity-dependent enzymatic labeling approaches in Figure S15. However, there is a question of whether there is a comparable uniformity in the measurement and description of this radius value across various articles. Most reviews consider the proximity labeling radius to be around 20nm. Therefore, the author is requested to provide a comparable benchmarking and then elucidate the distance constraint advantages of the CLASP method, which could more precisely define the interactions within subcellular spaces.

2)The use of internal conformation comparisons for DSSO-based CLASP effectively demonstrates that the distance constraint is essentially around 4nm. Please further incorporate structural information between subunits of protein complexes to substantiate the constrained distances of protein interactions, and provide illustrative examples in the main text to enhance the evidence for the labeling radius in the article.

3)If additional characterization could be provided in the method development section, such as such as super-resolution fluorescence spectroscopy or fluorescence resonance energy transfer (FRET) in the method establishment section to demonstrate the distance constraint radius of CLASP labeling, would greatly enhance the readability of the article. These techniques can provide visual and quantitative evidence of the proximity interactions and the spatial resolution that CLASP can achieve. By showing that the labeling occurs within a defined radius, it would help to validate the method's specificity and effectiveness in capturing protein-protein interactions within the complex cellular environment. This additional data would strengthen the readers' understanding of the CLASP technology and its potential applications in biological research.

4)It is recommended that the discussion section be expanded to include a more detailed comparison of the CLASP method with other proximity labeling techniques that are currently widely studied.

Proximity labeling methods with high specificity for live-cell in situ targeting have been extensively reported, and Alice Ting et al. have conducted a series of works on subcellular compartmentalization and membrane topology, demonstrating excellent performance.

In this section, a discussion could highlight the unique features and applications of CLASP in comparison to other proximity labeling methods. This could include a comparison of the labeling efficiency, specificity, temporal resolution, and the ability to study dynamic processes in living cells. Additionally, by discussing the complementary nature of crosslinking in the field of proximity labeling, readers can gain a better understanding of the importance of CLASP in the context of cellular biology and biochemistry. This discussion would also serve to illuminate the main themes of the article, showcasing the strengths and potential applications of CLASP as a tool for studying protein-protein interactions and the spatial organization of proteins within cells.

Reviewer #2 (Remarks to the Author):

Zhu et al. describe the use of cross-link assisted spatial proteomics (CLASP) to map suborganelle proteomes and membrane protein topology using mitochondria and synaptic vesicles (SV) as model systems.

The authors have addressed all my concerns especially relating to how the SV data was presented, and have performed several additional experiments addressing concerns raised by the other reviewers. I have no further concerns regarding this manuscript.

Reviewer #2 (Remarks on code availability):

I have looked at the info and the code on the Github page, but I have not tried to run the code myself, nor do I have the competence to judge whether the code is correct and functional.

Reviewer #3 (Remarks to the Author):

In the revised version of their manuscript, Zhu et al have sufficiently addressed all experimental concerns raised by this reviewer. This reviewer would only like to provide one additional comment on Figure S7: In this figure, the authors present overview pictures of cells while then performing Pearson correlation analyses for subcellular regions which are not further defined. It is only stated that they have been

manually selected. It would be good if the authors could include zoom-ins of individual cells to exemplify the manually selected regions mentioned.

Having said that, this reviewer still believes that the concept of CLASP does not present the novelty to justify publication in Nature Communications for the reasons outlined in the last round of review. As also mentioned by the authors, cross linking mass spectrometry has always been used to define protein-protein interactions and this practice does not provide a novel aspect as such. Utilizing well-known proteins to infer localization of others is further widely used across many approaches and does not provide novelty. While the authors argue that combination of both provides a high degree of novelty, this reviewer does not agree on that point. As also mentioned by Reviewer 4, these shortcomings could be compensated by additional novel biological insights through functional follow up experiments. Currently, however, the follow up experiments only focus on confirmation of the CLASPs ability to infer localization. Therefore, this reviewer stays with the initial assesement that a more specialized proteomics journal would be more appropriate.

Reviewer #4 (Remarks to the Author):

The authors have a provided an extensive and thoughtful rebuttal that both added new data and clarified the intent and utility of the CLASP method. I commend them on their efforts and feel that the manuscript is now well positioned for publication in Nature Communications.

REVIEWER COMMENTS

We thank all reviewers for taking the time to evaluate our revisions and their appreciation of this study.

Reviewer #1 (Remarks to the Author):

The author's revisions have greatly enhanced the quality of the article, yet there is a key issue that must be addressed before publication, as also highlighted by the comments of several other reviewers. The CLASP labeling radius is very important for determining sub-compartments, and within this paper, especially in the descriptions in the main text and the results figures, the emphasis on the argumentation should be increased, including:

As stated in our response to this reviewer's previous report, we agree that the CLASP labelling radius is an important element of this publication, which is why we followed this reviewer's suggestion to move the related Results from the Supplementary Information to the main text. Since neither this reviewer nor the other reviewers raised other concerns regarding our analysis of the CLASP labelling radius in their first reports, we are surprised that many new points are being brought up at this stage. Nonetheless, we will address them below.

1)The author refers to different literature reports regarding the labeling radius of proximity-dependent enzymatic labeling approaches in Figure S15. However, there is a question of whether there is a comparable uniformity in the measurement and description of this radius value across various articles. Most reviews consider the proximity labeling radius to be around 20nm. Therefore, the author is requested to provide a comparable benchmarking and then elucidate the distance constraint advantages of the CLASP method, which could more precisely define the interactions within subcellular spaces.

Indeed, many publications assume a proximity biotinylation labelling radius of 10-20 nm. To our knowledge, these publications adopted this number directly or indirectly from the work of Kim et al. (10.1073/pnas.1406459111) for BioID and Martell et al. (10.1038/nbt.2375) for APEX. These distances are shown in Supplementary Fig. 15 and both papers are cited in the corresponding figure legend. We are not aware of publications that independently confirmed this 10-20 nm radius. The additional papers cited in Supplementary Fig. 15 either determined labelling radii for variations of APEX/BioID or determined labelling radii by alternative approaches, which led to different numbers. As discussed in our response to this reviewer's previous report, all these numbers appeared in peer-reviewed journal articles, e.g. in PNAS. As such, we consider it most appropriate to present all these data to the readers in order to provide a full overview of the published information and give an indication of how much these numbers can vary between different studies (as also mentioned in the 4th paragraph of the Discussion).

We feel that a direct experimental comparison of the labelling radii of CLASP and other proximity biotinylation methods is beyond the scope of this paper, also considering that similar benchmarking studies have been published as stand-alone publications recently (e.g. Oakley

et al., PNAS 2022, 10.1073/pnas.2203027119). More importantly, though, we are not aware of any method that would allow a direct experimental comparison since the labelling radius of DSSO-based CLASP supersedes the spatial resolution of super-resolution microscopy assays (such as those used in the Oakley et al. benchmarking paper mentioned above) or FRET-based assays (>4.4 nm + the diameter of the fluorescent proteins that need to be tagged to the target proteins, see for example Table 3 in Bajar et al., Sensors 2016, 10.3390/s16091488)

While we are unable to ultimately ascertain the correctness of the previously published labelling radii of proximity biotinylation methods, we believe that the 4 nm labelling radius of DSSO-based CLASP we report is soundly supported by data. On the one hand, we provide several lines of supporting evidence in this paper including comparison of our cross-links to published PDB structures and independently established ultrastructural features of mitochondria. On the other hand, a maximum labelling radius of 4 nm for DSSO-based CLASP is well in line with previous DSSO cross-linking studies, which typically assume a maximum distance of 3-3.5 nm between the C α atoms of cross-linked lysine side chains (e.g. Wang et al., Mol Cell Proteom 2017, 10.1074/mcp.M116.065326 / Liu et al., Mol Cell Proteom 2018, 10.1074/mcp.RA117.000470 / Yu et al., Anal Chem 2022, 10.1021/acs.analchem.1c05298 / Jiao et al., Anal Chem 2022, 10.1021/acs.analchem.1c04485 / O'Reilly et al., Mol Syst Biol 2023, 10.15252/msb.202311544).

2)The use of internal conformation comparisons for DSSO-based CLASP effectively demonstrates that the distance constraint is essentially around 4nm. Please further incorporate structural information between subunits of protein complexes to substantiate the constrained distances of protein interactions, and provide illustrative examples in the main text to enhance the evidence for the labeling radius in the article.

The comparison of cross-links to PDB structures shown in Supplementary Figures 2A and 3 include both intra-links (cross-links within one protein) and inter-links (cross-links between different subunits of protein complexes).

Since similar structural comparisons have been published for DSSO cross-linking multiple times and have shown consistently that the DSSO distance constraint is ≤ 4 nm (see the references cited in our response to comment 1), we feel it is sufficient to present this analysis as Supplementary Figures.

Action taken: We have specified in the Supplementary Figure legends how many intra-links and inter-links are shown. We also clarify in the main text that both intra-links and inter-links have been mapped onto PDB structures. Furthermore, all mapped cross-links and their distances are included in the Source Data file.

3)If additional characterization could be provided in the method development section, such as such as super-resolution fluorescence spectroscopy or fluorescence resonance energy transfer (FRET) in the method establishment section to demonstrate the distance constraint radius of CLASP labeling, would greatly enhance the readability of the article. These techniques can provide visual and quantitative evidence of the proximity interactions and the

spatial resolution that CLASP can achieve. By showing that the labeling occurs within a defined radius, it would help to validate the method's specificity and effectiveness in capturing protein-protein interactions within the complex cellular environment. This additional data would strengthen the readers' understanding of the CLASP technology and its potential applications in biological research.

We agree that, in some scenarios, FRET can serve as a complementary technique to determine intermolecular distances. Unfortunately, as mentioned in our response to this reviewer's comment 1 above, the typical FRET labelling radii are larger than the maximum labelling radius of DSSO-based CLASP. In addition, FRET and CLASP labelling radii may not be directly comparable since CLASP directly measures the distance between the target proteins whereas FRET radii relate to the distance between the two fluorescent proteins conjugated to the target proteins.

4) It is recommended that the discussion section be expanded to include a more detailed comparison of the CLASP method with other proximity labeling techniques that are currently widely studied.

We agree that the comparison of CLASP and proximity biotinylation methods is critical for the evaluation of our approach, which is why our current discussion is focused on precisely this comparison. We comment on the reviewer's specific suggestions below.

Proximity labeling methods with high specificity for live-cell in situ targeting have been extensively reported,

We agree and this is already mentioned in the discussion ("this strategy has undisputed advantages for profiling cell type- or localization-specific proteomes in organisms⁵⁴⁻⁵⁶"),

and Alice Ting et al. have conducted a series of works on subcellular compartmentalization and membrane topology, demonstrating excellent performance.

We agree and have dedicated a separate Results sub-section to comparing CLASP to the APEX-based membrane topology work from Alice Ting lab alumni Hyun-Woo Rhee ("CLASP characterizes the topologies of membrane proteins"). The multiple application of Alice Ting's work and their difference to CLASP are then summarized in the discussion ("These studies demonstrated the capabilities of APEX-based proximity labeling to reveal protein sub-compartment localization^{27,30}, membrane protein topology²⁶, and membrane contact sites²⁵. Characterizing each of these spatial features required the design of dedicated proximity labeling experiments, whereas we have shown that CLASP can provide insights into all these aspects simultaneously in one experiment and from native mitochondria.").

In this section, a discussion could highlight the unique features and applications of CLASP in comparison to other proximity labeling methods.

Our Discussion currently includes paragraphs about the differences of CLASP and APEX/BioID experimental design (paragraph 2), data analysis (paragraph 3), and spatial resolution (paragraph 4) as well as a discussion of the application opportunities and limitations

of CLASP (final paragraph). In addition, as mentioned above, we highlight the unique advantages of APEX/BioID for in vivo applications in organisms. As such, we feel that this request is already addressed in the current Discussion.

This could include a comparison of the labeling efficiency, specificity, temporal resolution, and the ability to study dynamic processes in living cells.

Regarding labeling efficiency and specificity, we already discuss that CLASP increases the yield of spatial information per experiment (see 2nd paragraph Discussion) and yields a larger fraction of specific localization annotations (see 4th paragraph Discussion).

Temporal resolution – and the related ability to study dynamic cellular processes – is an interesting point, which we so far omitted from the discussion because, more than the other aspects, temporal resolution will vary greatly depending on the specific proximity labeling method or cross-linker used. As mentioned in our paper, CLASP is, in principle, compatible with any cross-linker. While DSSO and DSBSO cross-linking reactions as performed in this study are usually allowed to proceed for at least 15 min, other cross-linking chemistries with reaction times under 30 s have been published recently (e.g. Wang et al., Nat Commun 2022, 10.1038/s41467-022-28879-4) and may offer additional options for temporally resolved CLASP in the future.

For proximity labeling approaches, it is generally understood that BioID-based approaches offer lower temporal resolution than APEX-based approaches. However, for a fair discussion of this aspect, one would also have to consider other (especially light-driven) proximity labeling methods (e.g. Zheng et al. Nat Commun 2023, 10.1038/s41467-023-38565-8, and Zhai et al., Nat Commun 2022, <https://doi.org/10.1038/s41467-022-32689-z>). We did not discuss these methods in our manuscript because their spatial specificity on the sub-compartment and membrane topology levels – which are the focal points of our work – have not yet been evaluated.

Overall, we believe that doing the question of temporal resolution justice would substantively expand our manuscript, while remaining largely speculative since we did not characterize CLASP's temporal resolution so far. Given that our article is already quite long (Introduction+Results+Discussion = ca. 5700 words), we would prefer to refrain from an in-depth discussion of this point.

Action taken: We now mention in the Discussion that temporal resolution of CLASP and proximity labeling approaches depends on the choice of cross-linker and labeling method, respectively, citing the above-mentioned publications as examples. We also point out that the development of time-resolved CLASP is a subject for future studies.

Additionally, by discussing the complementary nature of crosslinking in the field of proximity labeling, readers can gain a better understanding of the importance of CLASP in the context of cellular biology and biochemistry. This discussion would also serve to illuminate the main themes of the article, showcasing the strengths and potential applications of CLASP as a tool for studying protein-protein interactions and the spatial organization of proteins within cells.

We fully agree with the scope this reviewer suggests for the Discussion; however, we are unsure what to change since our current Discussion is already written with the goal to cover exactly the aspects mentioned by the reviewer. To summarize:

- *We mention the advantages of APEX/BioID for in vivo applications and the limitations of CLASP in this regard (2nd paragraph and final paragraph of the Discussion)*
- *We discuss that CLASP's current primary application areas will be purifiable organelles and membrane-less compartments (final paragraph of the discussion)*
- *We mention the advantages of CLASP with regard to spatial resolution, sub-compartment specificity, and information yield per experiment (2nd and 4th paragraph of the Discussion)*
- *We explain the different data analysis principles of proximity biotinylation and CLASP experiments (3rd paragraph of the Discussion)*
- *We explain the complementary design of proximity biotinylation and CLASP experiments (2nd paragraph of the discussion) and propose how a synthesis of these design principles could benefit certain applications (6th paragraph of the discussion)*

We feel that these points together with additional text revisions proposed above cover all aspects this reviewer would like to see included in the revision.

Reviewer #2 (Remarks to the Author):

Zhu et al. describe the use of cross-link assisted spatial proteomics (CLASP) to map suborganelle proteomes and membrane protein topology using mitochondria and synaptic vesicles (SV) as model systems.

The authors have addressed all my concerns especially relating to how the SV data was presented, and have performed several additional experiments addressing concerns raised by the other reviewers. I have no further concerns regarding this manuscript.

Reviewer #2 (Remarks on code availability):

I have looked at the info and the code on the Github page, but I have not tried to run the code myself, nor do I have the competence to judge whether the code is correct and functional.

We thank this reviewer for the positive evaluation of our work.

Reviewer #3 (Remarks to the Author):

In the revised version of their manuscript, Zhu et al have sufficiently addressed all experimental concerns raised by this reviewer.

We thank this reviewer for appreciating our revisions.

This reviewer would only like to provide one additional comment on Figure S7: In this figure, the authors present overview pictures of cells while then performing Pearson correlation analyses for subcellular regions which are not further defined. It is only stated that they have been manually selected. It would be good if the authors could include zoom-ins of individual cells to exemplify the manually selected regions mentioned.

We apologize for not describing our approach more clearly in the legend. Co-localization analysis was performed for regions of interest within the overview images, i.e. regions of the image showing individual cells that could be manually masked out. We realize that the term “regions of interest” was ambiguous and have revised the figure legend. We also added example zoom-ins of masked-out cells.

Action taken: We include representative zoom-ins of masked out cells as Supplementary Fig 7B and have revised the figure legend to clarify that individual cells have been used for colocalization analysis.

Having said that, this reviewer still believes that the concept of CLASP does not present the novelty to justify publication in Nature Communications for the reasons outlined in the last round of review. As also mentioned by the authors, cross linking mass spectrometry has always been used to define protein-protein interactions and this practice does not provide a novel aspect as such. Utilizing well-known proteins to infer localization of others is further widely used across many approaches and does not provide novelty. While the authors argue that combination of both provides a high degree of novelty, this reviewer does not agree on that point. As also mentioned by Reviewer 4, these shortcomings could be compensated by additional novel biological insights through functional follow up experiments. Currently, however, the follow up experiments only focus on confirmation of the CLASPs ability to infer localization. Therefore, this reviewer stays with the initial assessment that a more specialized proteomics journal would be more appropriate.

We appreciate this reviewer’s opinion although we don’t share it. We certainly agree with the reviewer that XLMS has always been used to define protein-protein interactions. Complementary to this established XLMS application, we here provide a framework for using XLMS to infer protein localizations and membrane topologies, which – to our knowledge – has not been done before and constitutes the novelty of CLASP.

Reviewer #4 (Remarks to the Author):

The authors have provided an extensive and thoughtful rebuttal that both added new data and clarified the intent and utility of the CLASP method. I commend them on their efforts and feel that the manuscript is now well positioned for publication in Nature Communications.

We are glad that we could resolve this reviewer's concerns and thank them for their positive evaluation of our manuscript.